# Self-consistent theory of many-body localisation in a quantum spin chain with long-range interactions

Sthitadhi Roy[1,2,*] and David E. Logan[1,3]

**1** Physical and Theoretical Chemistry, Oxford University, South Parks Road, Oxford OX1 3QZ, United Kingdom
**2** Rudolf Peierls Centre for Theoretical Physics, Clarendon Laboratory, Oxford University, Parks Road, Oxford OX1 3PU, United Kingdom
**3** Department of Physics, Indian Institute of Science, Bangalore 560 012, India
* sthitadhi.roy@chem.ox.ac.uk

July 30, 2019

## Abstract

Many-body localisation is studied in a disordered quantum spin-1/2 chain with long-ranged power-law interactions, and distinct power-law exponents for interactions between longitudinal and transverse spin components. Using a self-consistent mean-field theory centring on the local propagator in Fock space and its associated self-energy, a localisation phase diagram is obtained as a function of the power-law exponents and the disorder strength of the random fields acting on longitudinal spin-components. Analytical results are corroborated using the well-studied and complementary numerical diagnostics of level statistics, entanglement entropy, and participation entropy, obtained via exact diagonalisation. We find that increasing the range of interactions between transverse spin components hinders localisation and enhances the critical disorder strength. In marked contrast, increasing the interaction range between longitudinal spin components is found to enhance localisation and lower the critical disorder.

# 1 Introduction

The presence of disorder in nature is as much an inevitability as it is a source of rich and often unexpected phenomena. In quantum condensed matter, much of the study of disordered systems falls under the umbrella of Anderson localisation, with its origins in Anderson's seminal work [1] showing that sufficiently strong disorder can induce spatial localisation of the wavefunctions of a system of non-interacting particles. In fact in one-dimension, Mott and Twose [2] later showed that single-particle states are localised even for an infinitesimally small disorder strength. A natural subsequent question is the robustness of localisation to the inclusion of interactions, the importance of which has long been appreciated and studied in the context of ground state phases [1,3]. More recently, the last decade or so has seen considerable attention given to this issue for highly excited quantum states at finite energy densities above the ground state, under the banner of *many-body localisation* [4–6] (see Refs. [7,8] for reviews and further references therein). Its fundamental importance stems in part from the fact that many-body localised systems fail to thermalise, and hence lie beyond the established norms of thermodynamic ensembles in statistical mechanics; allowing e.g. for the possibility of novel phenomena such as emergent integrability and unusual quantum order extending to arbitrary energy densities [9,10]. Rapid progress in experimental quantum simulators, and observation of many-body localisation in such experiments [11–13], has also spurred theoretical development.

The great majority of theoretical studies on many-body localisation have focussed on models with short-ranged interactions. In $d = 1$ spatial dimension, extensive numerical studies [14–18], phenomenological real-space renormalisation group formulations [19–26], approaches based on local and non-local propagators in Fock space [27,28], and treatments of classical percolation analogues on Fock space [29,30], have shown that there exists a finite critical disorder for the many-body localisation transition, although the precise nature of the transition remains an open question.

On the other hand, the current literature on many-body localisation in systems with power-law interactions paints a relatively pessimistic picture of the possibility of localisation. Arguments based on simple resonance counting and breakdown of the locator expansion have suggested that systems with interactions longer-ranged than $1/r^{2d}$ cannot host a many-body localised phase [31–35]; though such arguments can be debated on the grounds that simple resonance counting does not account for correlations in the Fock space (in both off-diagonal and diagonal matrix elements of the many-body Hamiltonian), and that the breakdown of the bare locator expansion does not itself guarantee the absence of localisation. Interestingly enough, experiments with trapped ions [12] and dipolar systems [36,37], where power-law interactions appear naturally, seem to suggest the presence of localised phases in regimes

where common lore would deem localisation impossible. In fact, arguments for the low-energy theory based on bosonisation [38] show that low-temperature many-body localisation is indeed possible in such long-ranged interacting systems. Nevertheless, the question of whether localisation persists at infinite temperatures – in other words for eigenstates in the middle of the spectrum – remains very much open. That is the question we seek to address in the present work.

In order to obtain an analytical, albeit approximate, understanding of the localisation phase diagram, we study the local propagators in Fock space within a self-consistent mean-field framework [28]. We focus in particular on the imaginary part of the associated self-energy and its distribution. Its typical value is expected to vanish with unit probability in the localised phase, but correspondingly to be non-zero in the delocalised phase, thereby signalling the phase transition. Free from the approximations underlying the mean-field theory, its essential predictions are corroborated using numerical results obtained from exact diagonalisation, for the ubiquitous diagnostics of level statistics, entanglement entropy, and participation entropy, and their finite-size scaling analyses.

The archetypal model for studying many-body localisation in one-dimensional short-ranged systems is a chain of spinless fermions with a disordered onsite potential, and nearest-neighbour hoppings and density-density interactions, which, via a Jordan-Wigner transformation, maps onto the random-field XXZ spin-1/2 chain. In this work we consider a long-ranged generalisation of the disordered XXZ chain described by the Hamiltonian

$$\mathcal{H} = \sum_i h_i \sigma_i^z + J_z \sum_{i>j} \frac{\sigma_i^z \sigma_j^z}{(i-j)^\beta} + J \sum_{i>j} \frac{1}{(i-j)^\alpha} (\sigma_i^x \sigma_j^x + \sigma_i^y \sigma_j^y), \tag{1}$$

where the $\sigma$'s are Pauli matrices for spins-1/2, and the $h_i \in [-W, W]$ describes the disordered fields (independent random variables for each site $i$). The model in Eq. (1) conserves total magnetisation, $M_z = \sum_{i=1}^N \sigma_i^z$, whence one can work independently in each $M_z$ sector. We chose to work in the $M_z = 0$ sector, which has the largest Fock-space dimension $N_{\mathcal{H}}(M_z = 0) = \binom{N}{N/2}$ and dominates the $2^N$-dimensional Fock space of all $M_z$ sectors in the thermodynamic limit (system size $N \to \infty$). The infinite temperature trace, whenever referred to henceforth, thus denotes the trace over all states in the $M_z = 0$ sector. Although we consider the $M_z = 0$ sector explicitly, we add that the analysis holds for all $M_z$ sectors whose Fock-space dimensions scale exponentially with $N$.

As these fields couple to the $\sigma^z$-component of the spins, we refer to the interaction between the $\sigma^z$ spin components, proportional to $J_z$ and decaying as a power law with exponent $\beta$ in the separation between the spins, as the *longitudinal interaction*. Similarly, we refer to the interaction between the $\sigma^x$ and $\sigma^y$ spin components, proportional to $J$ and decaying with an exponent $\alpha$, as the *transverse interaction*. The long-ranged interacting spin chain is not trivially related to a fermionic problem with long-ranged hopping, though we comment on the connection of our results to those of fermionic models later in the paper.

The central result of this work is the localisation phase diagram in the three-dimensional parameter space spanned by $\alpha$, $\beta$, and $W$. We find that making the transverse interactions longer ranged (by decreasing $\alpha$) aids delocalisation and increases the critical disorder strength for localisation .The mean-field treatment in fact predicts that the model Eq. (1) lacks a localised phase for $\alpha < 0.5$. On the other hand, quite remarkably, making the longitudinal interactions longer ranged (decreasing $\beta$) favours localisation and lowers the critical disorder. In fact the mean-field theory in this case predicts that the system is always localised for

$\beta < 0.5$. Physically, the long-ranged longitudinal interaction can be understood as providing the system with a rigidity against the spin-flips arising from transverse interactions, thus aiding localisation and eventually driving the system into a phase similar to an interaction-induced localised one.

The paper is organised is follows. In Sec. 2 we describe the local Fock-space propagators and their associated self-energies, discussing how the thermodynamic limit can be taken appropriately and how they act as indicators of the many-body localisation transition. The self-consistent calculation for the imaginary part of the self-energy is set up in Sec. 3, following the discussion in Ref. [28]. In Sec. 4 we employ the mean-field theory for the treatment of the long-ranged disordered XXZ chain Eq. (1) and derive the phase diagram of the model, which is then compared to numerical exact diagonalisation results in Sec. 5. We finally close with discussion and concluding remarks in Sec. 6.

## 2 Local Fock-space propagators and self-energies

The Hamiltonian of a generic quantum many-body system can always be expressed as a tight-binding Hamiltonian in Fock space,

$$\mathcal{H} = \sum_I \mathcal{E}_I \left| I \right\rangle \left\langle I \right| + \sum_{I \neq K} \mathcal{J}_{IK} \left| I \right\rangle \left\langle K \right|, \tag{2}$$

where $\{\left| I \right\rangle\}$ denotes a set of many-body basis states of the $N_\mathcal{H}$-dimensional Fock space, which act as the Fock-space sites of the tight-binding Hamiltonian Eq. (2). For a one-dimensional chain of spins-1/2 with disordered fields coupling to the $z$-component of the spins, a natural and convenient choice of the Fock space is the *configuration space*, with the sites $\left| I \right\rangle$ corresponding to product states in the basis of $\{\sigma_\ell^z\}$. One then expects eigenstates in the many-body localised phase to behave fundamentally differently on the Fock space from those of the delocalised phase. That this is indeed the case has been shown e.g. via numerical results for participation entropies and participation ratios [17, 39, 40]: in the many-body delocalised and localised phases respectively, eigenstates typically have support on $\mathcal{O}(N_\mathcal{H})$ and $\mathcal{O}(N_\mathcal{H}^\alpha)$; $\alpha < 1$ Fock space sites, which are respectively finite and vanishing fractions of the Fock-space dimension.

Collating the above two aspects of the problem of many-body localisation, propagators in Fock space seem natural quantities to consider, since their real-space single-particle analogues have long been profitably studied in problems of Anderson localisation [1, 41]. It is important to realise that there exist fundamental differences between the problem of many-body localisation recast as a disordered tight-binding model on a high-dimensional graph, and single-particle localisation problems in high dimensions; and considerable caution needs to be exercised in invoking understandings from high-dimensional Anderson localisation. Fortunately these issues are not insurmountable, inasmuch as there have been recent works which have used a mean-field treatment of the local Fock-space propagator [28], as well their non-local counterparts within the forward scattering approximation [27], to understand the many-body localisation transition.

We will concern ourselves exclusively with the local Fock-space propagator

$$G_I(t) = -i\Theta(t) \left\langle I \right| e^{-i\mathcal{H}t} \left| I \right\rangle \xleftrightarrow{\ G_I(\omega) = \int dt\ G_I(t) e^{i\omega^+ t}\ } G_I(\omega) = \left\langle I \right| (\omega^+ - \mathcal{H})^{-1} \left| I \right\rangle, \tag{3}$$

the Lehmann representation of which is

$$G_I(\omega) = \sum_{n=1}^{N_{\mathcal{H}}} \frac{|A_{nI}|^2}{\omega^+ - E_n}. \tag{4}$$

Here, $A_{nI} = \langle I|\psi_n \rangle$ with $|\psi_n\rangle$ an eigenstate of $\mathcal{H}$ with eigenvalue $E_n$, and $\omega^+ = \omega + i\eta$ with $\eta = 0^+$. The local propagator is of particular importance as it provides access to two classic probes of localisation, the local density of states and the imaginary part of the self-energy [41–43]. While these have been used extensively in studying single-particle localisation, crucial differences arise in the context of many-body localisation. We now describe briefly the two notions, taking care to emphasise these important differences, especially in regard to taking the thermodynamic limit. As shown below, this motivates a necessary rescaling of the energy scales of the problem in the many-body case, such that the local density of states and imaginary part of the self-energy have well-defined thermodynamic limits [28].

The local density of states follows from $G_I(\omega)$ as

$$D_I(\omega) = -\frac{1}{\pi}\mathrm{Im}G_I(\omega) = \sum_n |A_{nI}|^2 \delta(\omega - E_n) \tag{5}$$

(and is normalised to unity over $\omega$). Physically, $D_I(\omega)$ is a measure of the number of eigenstates of energy $\omega$ which overlap Fock-space site $I$. In the context of single-particle localisation, it is well known that $D_I(\omega)$ (with $I$ in this case denoting real-space sites) is pure point-like in the localised phase, and absolutely continuous in the delocalised phase. This reflects the fact that, due to exponential localisation (in real-space) of states in the former case, only a finite number $\mathcal{O}(1)$ of eigenstates with energies close to $\omega$ can overlap any real-space site; while in the delocalised phase by contrast, that number is proportional to the system size, and hence $D_I(\omega)$ forms a continuum in the thermodynamic limit. The situation is slightly more delicate in the case of many-body localisation, where the spectrum $D_I(\omega)$ strictly speaking forms a continuum in the thermodynamic limit in both phases. However, the number of eigenstates close to some given energy $\omega$ which overlap a Fock-space site $I$ are, respectively, vanishing and finite *fractions* of the Fock-space dimension in the localised and delocalised phases in the thermodynamic limit (similarly, the ratio of the number of Fock-space sites on which an eigenstate has support in the MBL phase, to the corresponding number in the delocalised phase, vanishes in the thermodynamic limit, as implied by the behaviour of participation entropies [17, 39, 40]). This suggests that the spectrum of $D_I(\omega)$ will appear point-like in a many-body localised phase if viewed on energy scales relative to that for the delocalised phase.

An essential characteristic of the many-body delocalised phase can in turn be understood by considering the limit of weak disorder, under the standard assumption made here that all basis states are essentially equivalent and hence $|A_{nI}|^2 \sim 1/N_{\mathcal{H}}$. Eq. (5) then gives $D_I(\omega) \simeq N_{\mathcal{H}}^{-1}\sum_n \delta(\omega - E_n) = D(\omega)$, with $D(\omega)$ the normalised *total* density of eigenstates. Since one expects the latter to be Gaussian [44], $D_I(\omega) \propto \mu_E^{-1}$, with $\mu_E$ the standard deviation/width of the total density of states. But for generic many-body systems $\mu_E$ diverges in the thermodynamic limit $N \to \infty$ (with $\mu_E \propto \sqrt{N}$ for short-ranged models [44]). Hence the appropriate quantity to consider is the rescaled local density of states, $\tilde{D}_I = \mu_E D_I$. With this rescaling of the local spectrum (and hence propagator), the thermodynamic limit can safely be taken. This is a first indication that the energy scales in the problem should be rescaled with $\mu_E$.

We turn our attention next to the self-energy, $\Sigma_I(\omega)$, defined via the local propagator as

$$G_I(\omega) = [\omega^+ - \mathcal{E}_I - \Sigma_I(\omega)]^{-1}; \quad \Sigma_I(\omega) = X_I(\omega) - i\Delta_I(\omega), \tag{6}$$

where $X_I(\omega)$ and $\Delta_I(\omega)$ respectively denote its real and imaginary parts; we will be particularly interested in the latter. Physically, $\Delta_I(\omega)$ can be interpreted as the inverse lifetime associated with the decay of weight from $|I\rangle$ into states of energy $\omega$, and hence it naturally acts as a diagnostic for a localisation transition. In the context of single-particle localisation for example, it is well understood that in the localised phase $\Delta_I(\omega)$ is vanishingly small with unit probability over an ensemble of disorder realisations; specifically $\Delta_I(\omega) \propto \eta \to 0^+$. In a delocalised phase by contrast, $\Delta_I(\omega)$ is non-zero and finite with probability unity.

As with the local density of states, caution must however be exercised in taking the thermodynamic limit [28]. From the definition of the self-energy in Eq. (6), one can express $\Delta_I(\omega)$ as

$$\Delta_I(\omega) = \frac{\pi D_I(\omega)}{\text{Re}[G_I(\omega)]^2 + [\pi D_I(\omega)]^2} - \eta, \tag{7}$$

where the Lehmann representation of $G_I(\omega)$, Eq. (4), gives

$$\text{Re}[G_I(\omega)] = \sum_{n=1}^{N_{\mathcal{H}}} \frac{(\omega - E_n)|A_{nI}|^2}{(\omega - E_n)^2 + \eta^2}, \qquad \pi D_I(\omega) = \sum_{n=1}^{N_{\mathcal{H}}} \frac{\eta|A_{nI}|^2}{(\omega - E_n)^2 + \eta^2}. \tag{8}$$

As pointed out above, deep in the delocalised phase $|A_{nI}|^2 \sim N_{\mathcal{H}}^{-1}$, and hence $D_I(\omega) \simeq D(\omega)$ with $D(\omega)$ the total density of states. Since $\text{Re}[G_I(\omega)]$ is related to its spectral density $D_I(\omega)$ by a Hilbert transform, it follows likewise that $\text{Re}[G_I(\omega)] \simeq \text{Re}[G(\omega)]$ (the Hilbert transform of $D(\omega)$). The important point here is that $D(\omega)$, and hence $\text{Re}[G(\omega)]$, are each proportional to $\mu_E^{-1}$. From Eq. (7) it follows immediately that $\Delta_I(\omega) \propto \mu_E$ in the many-body delocalised phase. And since $\mu_E$ itself diverges as $N \to \infty$, it is thus $\tilde{\Delta}_I = \Delta_I/\mu_E$ that admits a well-defined thermodynamic limit, and as such is the appropriate quantity to study. Here we have of course shown this explicitly in the weak-disorder regime, but the result holds in general throughout the delocalised phase.

The essential message of this section was simply to point out that, to enable the thermodynamic limit to be taken, the relevant energy scales in the problem must be rescaled with the width of the density of eigenstates, and that quantities such as the appropriately rescaled self-energies or local densities of states are useful to study in the context of many-body localisation.

## 3 Imaginary part of the self-energy: self-consistent calculation

We now set up a self-consistent mean-field calculation for the appropriately rescaled imaginary part of the self-energy. The basic structure of the theory is the same as for the short-ranged case discussed in Ref. [28], where further information may be found.

Using the Feenberg renormalised perturbation series [42,43], the self-energy $\Sigma_I(\omega)$ can be expressed as

$$\Sigma_I(\omega) = \sum_K \mathcal{J}_{IK}^2 G_K(\omega) + \cdots$$

$$= \sum_K \frac{\mathcal{J}_{IK}^2}{\omega^+ - \mathcal{E}_K - \Sigma_K(\omega)} + \cdots . \tag{9}$$

Specifically, we consider the problem at the second order renormalised level only and neglect the higher order terms. In addition, as motivated and argued for in the previous section, all energies are rescaled with the standard deviation $\mu_E$ of the density of eigenstates. We thus consider $\tilde{\Sigma}_I = \Sigma_I/\mu_E$ in terms of $\tilde{G} = \mu_E G$, $\tilde{\omega} = (\omega - \overline{\mathcal{E}})/\mu_E$, and $\tilde{\mathcal{E}}_K = (\mathcal{E}_K - \overline{\mathcal{E}})/\mu_E$. Note that in addition to rescaling by $\mu_E$, the Fock-space site energies are taken relative to their mean $(\overline{\mathcal{E}})$, so that $\tilde{\omega} = 0$ corresponds to energies at the band centre where the density of states has a peak. With this, the rescaled self-energy can be expressed as

$$\tilde{\Sigma}_I(\tilde{\omega}) = \frac{1}{\mu_E^2} \sum_K \mathcal{J}_{IK}^2 \tilde{G}_K(\tilde{\omega}) = \frac{1}{\mu_E^2} \sum_K \frac{\mathcal{J}_{IK}^2}{\tilde{\omega}^+ - \tilde{\mathcal{E}}_K - \tilde{\Sigma}_K(\tilde{\omega})} \tag{10}$$

where $\tilde{\omega}^+ = \tilde{\omega} + i\tilde{\eta}$ with $\tilde{\eta} = \eta/\mu_E = 0^+$. This is now in a form which makes it amenable to a probabilistic mean-field treatment, consisting of three essential steps. The first consists of replacing the self-energy on the right-hand side of Eq. (10) by a typical value, $\tilde{\Sigma}_K(\tilde{\omega}) \to \tilde{\Sigma}_{\text{typ}}(\tilde{\omega}) = \tilde{X}_{\text{typ}}(\tilde{\omega}) - i\tilde{\Delta}_{\text{typ}}(\tilde{\omega})$. The second step is to obtain the probability distribution $P_{\tilde{\Delta}}(\tilde{\Delta}_I)$ for the imaginary part of the self-energy (at the chosen $\tilde{\omega}$), which itself depends on the typical value $\tilde{\Delta}_{\text{typ}}$. Finally, self-consistency is imposed by equating the 'input' $\tilde{\Delta}_{\text{typ}}$ to the typical value obtained from the geometric mean of the full distribution, as $\tilde{\Delta}_{\text{typ}} = \exp\left[\int_0^\infty d\tilde{\Delta}_I \, (\log \tilde{\Delta}_I) P_{\tilde{\Delta}}(\tilde{\Delta}_I)\right]$.

To proceed further on a concrete footing, we need to recast the Hamiltonian of the long-range interacting quantum spin chain, Eq. (1), in terms of the tight-binding Hamiltonian on the Fock space, Eq. (2), using a suitable choice of basis. Since disorder in the model couples to the $z$-component of the spins, the set of product states $|\{\sigma_l^z\}\rangle$ in the $z$-direction is a natural choice of basis, as they are eigenstates of the Hamiltonian in the $J = 0$ and infinite disorder limits. With this basis choice, the diagonal $(\{\mathcal{E}_I\})$ and off-diagonal $(\{\mathcal{J}_{IK}\})$ elements of the Fock-space tight-binding Hamiltonian can be identified as

$$\mathcal{E}_I = \langle I| \sum_{i>j} \frac{J_z}{(i-j)^\beta} \sigma_i^z \sigma_j^z + \sum_i h_i \sigma_i^z |I\rangle$$

$$\mathcal{J}_{IK} = \langle I| \sum_{i>j} \frac{J}{(i-j)^\alpha} (\sigma_i^x \sigma_j^x + \sigma_i^y \sigma_j^y) |K\rangle \tag{11}$$

(where $(\sigma_i^x \sigma_j^x + \sigma_i^y \sigma_j^y) = 2(\sigma_i^+ \sigma_j^- + \sigma_i^- \sigma_j^+)$ for the Pauli matrices we employ).

Inspection of Eq. (11) leads to the important observation that any pair of Fock-space basis states $|I\rangle$ and $|K\rangle$ with a finite $\mathcal{J}_{IK}$ differ only by a pair of spin-flips; and hence

$$|\mathcal{E}_I - \mathcal{E}_K| \sim \mathcal{O}(W, J_z) \quad \forall (I, K) \text{ such that } \mathcal{J}_{IK} \neq 0, \tag{12}$$

which naturally implies that for such pairs $|\tilde{\mathcal{E}}_I - \tilde{\mathcal{E}}_K| = |\mathcal{E}_I - \mathcal{E}_K|/\mu_E$ vanishes in the thermodynamic limit, since $\mu_E$ diverges. This is a manifestation of the fact that the on-site energies in the Fock-space tight-binding Hamiltonian are correlated, which makes this problem fundamentally different from Anderson localisation on high-dimensional graphs; in addition to the fact that the normalised density of states in the many-body problem scales with system size, unlike in a one-body problem. Within the probabilisitc mean-field framework, the self-energy in Eq. (10) can then be expressed as

$$\tilde{\Sigma}_I(\tilde{\omega}) = \frac{\Gamma_I^2}{\tilde{\omega}^+ - \tilde{\mathcal{E}}_I - \tilde{\Sigma}_{\text{typ}}(\tilde{\omega})}. \tag{13}$$

Here $\Gamma_I^2 = \sum_K \mathcal{J}_{IK}^2/\mu_E^2$, which encodes information about the connectivity of the state $|I\rangle$ on the Fock space weighted by the power-law decay of the interactions in Eq. (11), and hence depends on the power-law exponent $\alpha$. We will replace $\Gamma_I^2$ by its mean over the Fock-space graph, $\overline{\Gamma^2}$, with which the imaginary part of the rescaled self-energy reads

$$\tilde{\Delta}_I(\tilde{\omega}) = \frac{\overline{\Gamma^2}(\tilde{\eta} + \tilde{\Delta}_{\mathrm{typ}}(\tilde{\omega}))}{(\overline{\omega} - \tilde{\mathcal{E}}_I)^2 + (\tilde{\eta} + \tilde{\Delta}_{\mathrm{typ}}(\tilde{\omega}))^2} \tag{14}$$

where the real part of the self-energy has been absorbed for convenience into $\overline{\omega} := \tilde{\omega} - \tilde{X}_{\mathrm{typ}}(\tilde{\omega})$. It is clear from Eq. (14) that two ingredients are necessary to construct the probability distribution of $\tilde{\Delta}_I$: (i) the weighted average connectivity on Fock space, $\overline{\Gamma^2}$, and (ii) the distribution of the Fock-space site energies, which we denote by $P_{\tilde{\mathcal{E}}}$. Derivation of analytical expressions for the two, as functions of the power-law exponents $\alpha$ and $\beta$ respectively, will be focus of the next two subsections.

## 3.1   Average weighted connectivity on Fock space

Note from Eq. (11) that the off-diagonal part of the many-body Hamiltonian connects two Fock-space basis states by flipping a pair of anti-parallel spins at arbitrary separation, the corresponding matrix element being suppressed algebraically in the separation. The average weighted connectivity can thus be obtained by first calculating the average number of states, $\overline{Z(r)}$, to which any Fock-space basis state is connected by such a flip for a pair of spins separated by $r$, and then summing over all possible values of $r$ weighted with the corresponding matrix element. In the $M_z = 0$ sector considered, it is readily shown that the average connectivity corresponding to a pair of spin-flips at separation $r$ is

$$\overline{Z(r)} = \frac{1}{2} \frac{N}{(N-1)}(N - r) \tag{15}$$

Hence $\overline{\Gamma^2}$ can be calculated,

$$\overline{\Gamma^2} = \left(\frac{2J}{\mu_E}\right)^2 \sum_{r=1}^{N-1} \frac{\overline{Z(r)}}{r^{2\alpha}} \overset{N \gg 1}{=} \frac{2J^2}{\mu_E^2} \begin{cases} N\zeta(2\alpha); & \alpha > 1/2 \\ N \log N; & \alpha = 1/2 \\ z(\alpha)N^{2-2\alpha}; & \alpha < 1/2, \end{cases} \tag{16}$$

where the right-hand side gives the leading large-$N$ asymptotic behaviour of the sum. $\zeta(s)$ denotes the Riemann zeta function, and the function $z(\alpha)$ can be obtained by performing the summation in Eq. (16) exactly (modulo these prefactors, the leading large-$N$ form can in fact be obtained simply by replacing the sum in eq. (16) by an integral).

## 3.2   Moments of distributions of Fock-space site energies

We now turn our attention to the distribution of Fock-space site energies, $P_{\mathcal{E}}$. For the short-ranged limit of the model, with nearest-neighbour spin couplings ($\alpha = \infty = \beta$), it is known that $P_{\mathcal{E}}$ is precisely a Normal distribution [44], and thus characterised solely by its mean ($\overline{\mathcal{E}}$) and standard deviation ($\mu_{\mathcal{E}}$). We assume the same to hold for the long-ranged case. This is well justified by numerical results, which also corroborate the scalings of the mean and standard deviation with $N$ which we derive analytically below. Fig. 1 shows numerical results

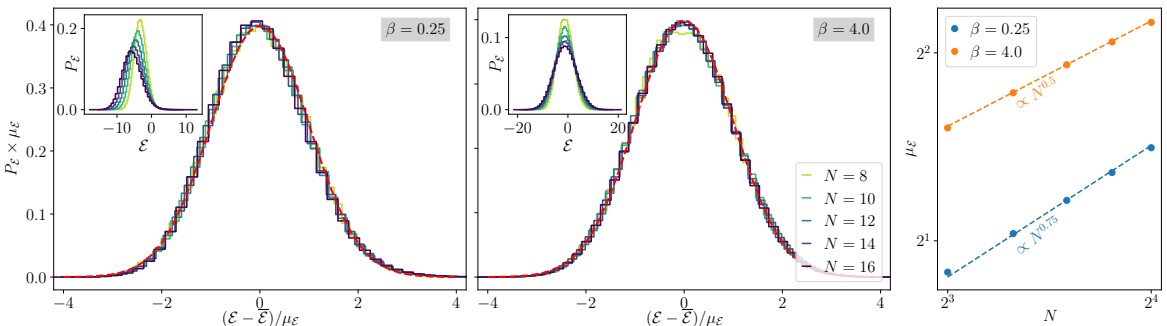

Figure 1: **Distributions of Fock-space site energies:** For $\beta = 0.25$ and 4, and for system sizes $N = 8 - 16$, the first two panels show the distributions $P_\mathcal{E}$ $vs$ $(\mathcal{E} - \overline{\mathcal{E}})/\mu_\mathcal{E}$, i.e. taken relative to their means and scaled with their standard deviations, $\mu_\mathcal{E}$. The red dashed line shows a standard Normal distribution, which clearly captures the numerics. The insets show the bare distributions, $vs$ $\mathcal{E}$. The right panel shows $\mu_\mathcal{E}$ $vs$ $N$ on logarithmic axes for the same two values of $\beta$. The exponents for the polynomial growth in $N$, as shown by the dashed lines, corroborate the predictions of Eq. (26). Results are shown for $J_z = 1 = W$.

for $P_\mathcal{E}$ for system sizes $N = 8 - 16$, and for two different values of $\beta$. In both cases, when the distributions are taken relative to their means and scaled with their standard deviations, they clearly collapse onto a common form for different system sizes. That common form is practically indistinguishable from a standard Normal distribution, shown by the red dashed line. From here on, we thus focus solely on the first two moments of $P_\mathcal{E}$.

We start with the first moment, $\overline{\mathcal{E}}$, which is given simply by

$$\overline{\mathcal{E}} = \left\langle \mathrm{Tr}' \left[ \sum_{i>j} \frac{J_z}{(i-j)^\beta} \sigma_i^z \sigma_j^z + \sum_i h_i \sigma_i^z \right] \right\rangle_{\mathrm{disorder}}. \tag{17}$$

Here $\mathrm{Tr}'[\cdot] = \sum'_I \langle I| \cdot |I\rangle / N_\mathcal{H}$, with the primed summation running over all Fock-space basis states satisfying $M_z = 0$, and the dimension of the corresponding Hilbert space is $N_\mathcal{H} = \binom{N}{N/2}$ for a system with $N$ spins. Using the result that in the $M_z = 0$ sector

$$\mathrm{Tr}'[\sigma_i^z] = 0 \quad \text{and} \quad \mathrm{Tr}'[\sigma_i^z \sigma_j^z] = -\frac{1}{N-1}, \tag{18}$$

$\overline{\mathcal{E}}$ can be expressed as

$$\overline{\mathcal{E}} = \frac{-J_z}{N-1} \sum_{i>j} \frac{1}{(i-j)^\beta} = \frac{-J_z}{N-1} \sum_{r=1}^{N-1} \frac{N-r}{r^\beta}, \tag{19}$$

where the second equality reflects the fact that the number of ways of finding a pair $i > j$ such that $i - j = r$ is $N - r$. The asymptotic behaviour of $\overline{\mathcal{E}}$ as the thermodynamic limit is approached is again readily obtained, with the limiting large-$N$ behaviour given for various ranges of $\beta$ by

$$\overline{\mathcal{E}} \overset{N \geqq 1}{=} \begin{cases} -J_z \zeta(\beta); & \beta > 1 \\ -J_z \log N; & \beta = 1 \\ -J_z y_1(\beta) N^{1-\beta}; & \beta < 1, \end{cases} \tag{20}$$

where $y_1$ is a function solely of $\beta$ that can be obtained from evaluating the summation in Eq. (19) exactly.

The second moment of the distribution can likewise be computed via

$$\overline{\mathcal{E}^2} = \left\langle \mathrm{Tr}' \left[ \left( \sum_{i>j} \frac{J_z}{(i-j)^\beta} \sigma_i^z \sigma_j^z + \sum_i h_i \sigma_i^z \right)^2 \right] \right\rangle_{\text{disorder}}. \tag{21}$$

The calculation is however slightly tedious, so we simply sketch the derivation here and relegate the details to Appendix A. To derive the expression for $\overline{\mathcal{E}^2}$, in addition to Eq. (18), we will use that in the $M_z = 0$ sector

$$\mathrm{Tr}'[\sigma_i^z \sigma_j^z \sigma_k^z] = 0 \quad \text{and} \quad \mathrm{Tr}'[\sigma_i^z \sigma_j^z \sigma_k^z \sigma_l^z] = \frac{3}{(N-1)(N-3)}, \tag{22}$$

for $i \neq j \neq k \neq l$. Using Eqs. (18) and (22) together with the fact that $\langle h_i \rangle_{\text{disorder}} = 0$ and $\langle h_i h_j \rangle_{\text{disorder}} = \delta_{ij} W^2/3$, Eq. (21) can be recast as

$$\overline{\mathcal{E}^2} = J_z^2 \left[ \frac{3}{(N-1)(N-3)} \Upsilon_0 - \frac{1}{N-1} \Upsilon_1 + \Upsilon_2 \right] + N \frac{W^2}{3}, \tag{23}$$

where

$$\Upsilon_0 = \sum_{i \neq j \neq k \neq l} \frac{1}{4|i-j|^\beta |k-l|^\beta}, \quad \Upsilon_1 = \sum_{i \neq j \neq k} \frac{1}{|i-j|^\beta |j-k|^\beta}, \quad \Upsilon_2 = \sum_{i \neq j} \frac{1}{2|i-j|^{2\beta}}. \tag{24}$$

Note from Eq. (21) that the terms proportional to $J_z^2$ will generically contain a product of four $\sigma^z$-operators. Physically, the $\Upsilon_0$ term is associated with the sum of such terms in which all four operators act on distinct sites, whence the prefactor to $\Upsilon_0$ in Eq. (23) is given by Eq. (22). Likewise, the $\Upsilon_1$ term corresponds to terms where only three of the four sites are distinct, i.e. of form $\sigma_i^z \sigma_j^z \sigma_k^z \sigma_k^z$. Since $[\sigma_k^z]^2 = 1$, we are left with $\mathrm{Tr}'$ of a product of two distinct $\sigma^z$-operators, and hence the prefactor is $-1/(N-1)$ (Eq. (18)). Finally, $\Upsilon_2$ corresponds to the sum of terms where just two of the site indices are unique, whence the overall operator squares to the identity, as reflected by the unit prefactor in Eq. (23).

As discussed in Appendix A, the leading large-$N$ asymptotic forms of the sums in Eq. (24) can be obtained, giving the asymptotic behaviour of $\overline{\mathcal{E}^2}$ as

$$\overline{\mathcal{E}^2} \stackrel{N \gg 1}{=} \begin{cases} (J_z^2 \zeta(2\beta) + W^2/3)N; & \beta > 1/2 \\ J_z^2 N \log N; & \beta = 1/2 \\ J_z^2 y_2(\beta) N^{2-2\beta}; & \beta < 1/2 \end{cases}. \tag{25}$$

where $y_2 = y_2(\beta)$ can be obtained exactly by evaluating the summations in Eq. (24).

With the large-$N$ forms of $\overline{\mathcal{E}}$ and $\overline{\mathcal{E}^2}$ at hand, the asymptotic behaviour of the standard deviation $\mu_{\mathcal{E}} = [\overline{\mathcal{E}^2} - \overline{\mathcal{E}}^2]^{1/2}$ in various ranges of $\beta$ can then be expressed as

$$\mu_{\mathcal{E}} \stackrel{N \gg 1}{=} \begin{cases} \sqrt{[J_z^2 \zeta(2\beta) + W^2/3] N}; & \beta > 1/2 \\ J_z \sqrt{N \log N}; & \beta = 1/2 \\ J_z \sqrt{(y_2(\beta) - y_1^2(\beta))} N^{1-\beta}; & \beta < 1/2 \end{cases}. \tag{26}$$

Three comments may be made here. First, the $N$-dependence of $\mu_{\mathcal{E}}$ in Eq. (26) is nicely exemplified by the numerical results of Fig. 1, right panel, where examples for both $\beta > 1/2$ and $< 1/2$ are shown. Second, for $\beta \leq 1/2$ the 'external' disorder strength $W$ arising from the disordered fields drops out of the leading asymptotics, because its $N$-dependence ($\propto N$) is sub-dominant to that arising from the spin-interaction contribution embodied in $J_z$. In physical terms, the occurrence of the latter reflects the fact that interactions effectively self-generate disorder in the $\{\mathcal{E}_I\}$, due to configurational disorder in the distribution of spins $\{\sigma_l^z\}$ prescribing the $|I\rangle$'s. Finally here, though essentially superfluous in the following, we mention for completeness that the variance $\mu_E^2$ of the density of states is readily obtained on noting that $\langle I|H^2|I\rangle = \mathcal{E}_I^2 + \sum_K \mathcal{J}_{IK}^2$, and is given by

$$\mu_E^2 = \mu_{\mathcal{E}}^2 + \overline{\sum_K \mathcal{J}_{IK}^2} \;\equiv\; \mu_{\mathcal{E}}^2 + \mu_E^2 \overline{\Gamma^2} \tag{27}$$

where the leading $N$-dependence of $\mu_E^2 \overline{\Gamma^2}$ is given explicitly by Eq. (16).

As shown in subsequent sections, it is the scaling of $\overline{\Gamma^2}$ and $\mu_{\mathcal{E}}$ with system size $N$, Eqs. (16) and (26) respectively, which play a crucial role in determining the phase diagram of the model in the $(\alpha, \beta)$ parameter space.

### 3.3 Criterion for the many-body localisation transition

Having established that $P_{\mathcal{E}}$ is normally distributed, and obtained explicit expressions for its moments as well as for $\overline{\Gamma^2}$, we can self-consistently compute the distribution of $\tilde{\Delta}_I$ using Eq. (14) as

$$P_{\tilde{\Delta}}(\tilde{\Delta}) = \int_{-\infty}^{\infty} d\tilde{\mathcal{E}}_I \, P_{\tilde{\mathcal{E}}}(\tilde{\mathcal{E}}_I) \, \delta\left(\tilde{\Delta} - \frac{\overline{\Gamma^2}(\tilde{\eta} + \tilde{\Delta}_{\text{typ}}(\tilde{\omega}))}{(\overline{\omega} - \tilde{\mathcal{E}}_I)^2 + (\tilde{\eta} + \tilde{\Delta}_{\text{typ}}(\tilde{\omega}))^2}\right), \tag{28}$$

where

$$P_{\tilde{\mathcal{E}}}(\tilde{\mathcal{E}}_I) = \frac{1}{\sqrt{2\pi\mu_{\tilde{\mathcal{E}}}^2}} \exp\left(-\frac{\tilde{\mathcal{E}}_I^2}{2\mu_{\tilde{\mathcal{E}}}^2}\right) \qquad : \; \mu_{\tilde{\mathcal{E}}} = \mu_{\mathcal{E}}/\mu_E. \tag{29}$$

The Normal form of $P_{\tilde{\mathcal{E}}}$ allows us to do the integration in Eq. (28) analytically, yielding

$$P_{\tilde{\Delta}}(\tilde{\Delta}) = \left[1 - \frac{\tilde{\Delta}(\tilde{\eta} + \tilde{\Delta}_{\text{typ}})}{\overline{\Gamma^2}}\right]^{-1/2} \sqrt{\frac{\kappa}{\pi}} \frac{1}{\tilde{\Delta}^{3/2}} \exp\left[-\kappa\left(\frac{1}{\tilde{\Delta}} - \frac{\tilde{\eta} + \tilde{\Delta}_{\text{typ}}}{\overline{\Gamma^2}}\right)\right] \tag{30}$$

where $\kappa = \overline{\Gamma^2}(\tilde{\eta} + \tilde{\Delta}_{\text{typ}})/2\mu_{\tilde{\mathcal{E}}}^2$, and we set $\overline{\omega} = 0$ (equivalently $\tilde{\omega} = 0$), which corresponds to band centre states of energy $\omega = \text{Tr}'[\mathcal{H}]$. Self-consistency can then be imposed by calculating the typical value of $\tilde{\Delta}$ from this distribution and equating it to $\tilde{\Delta}_{\text{typ}}$, via

$$\exp\left[\int_0^{\infty} d\tilde{\Delta} \, P_{\tilde{\Delta}}(\tilde{\Delta}) \log \tilde{\Delta}\right] = \tilde{\Delta}_{\text{typ}}. \tag{31}$$

In the following we impose self-consistency separately in the two phases, as done in Ref. [28] for a short-ranged system. The criterion for each of the two phases to exist self-consistently is found to break down at the same point in parameter space, indicating that the point (or set of such points) is a critical point for the many-body localisation transition.

We start with the localised phase, in which $\Delta_{\text{typ}} \propto \eta$ is vanishingly small and hence the appropriate distribution to study is that of $y := \Delta/\eta = \tilde{\Delta}/\tilde{\eta}$. Since $\tilde{\eta} \to 0$, the distribution for $y$ follows directly from Eq. (30) as

$$P_y(y) = \sqrt{\frac{\kappa}{\tilde{\eta}\pi}} \, \frac{1}{y^{3/2}} \exp\left(-\frac{\kappa}{\tilde{\eta}y}\right) \qquad : \; y = \frac{\tilde{\Delta}}{\tilde{\eta}}, \tag{32}$$

which is precisely a normalised Lévy distribution (with the expected power-law tail [28] $\propto y^{-3/2}$). Hence $\tilde{\Delta}_{\text{typ}}$ can be computed as

$$\int_0^\infty dy \, P_y(y) \log y \;=\; \log\left(4\kappa/\tilde{\eta}\right) + \gamma \;=\; \log\left(\tilde{\Delta}_{\text{typ}}/\tilde{\eta}\right), \tag{33}$$

where $\gamma \, (= 0.577216..)$ is the Euler-Mascheroni constant. Since $\kappa = \overline{\Gamma^2}(\tilde{\eta} + \tilde{\Delta}_{\text{typ}})/2\mu_{\tilde{\mathcal{E}}}^2$, solution of this self-consistency condition for $\tilde{\Delta}_{\text{typ}}/\tilde{\eta}$ yields

$$\frac{\tilde{\Delta}_{\text{typ}}}{\tilde{\eta}} = \frac{2\overline{\Gamma^2}}{\mu_{\tilde{\mathcal{E}}}^2} e^\gamma \left(1 - \frac{2\overline{\Gamma^2}}{\mu_{\tilde{\mathcal{E}}}^2} e^\gamma\right)^{-1}. \tag{34}$$

Recall that in physical terms $\tilde{\Delta}$ is effectively an inverse lifetime, and is thus non-negative. Hence from Eq. (34), the many-body localised phase is self-consistently possible only if

$$\Lambda := \frac{2\overline{\Gamma^2}}{\mu_{\tilde{\mathcal{E}}}^2} e^\gamma \leq 1, \tag{35}$$

where the equality corresponds to points in parameter space which give the limits of stability of the self-consistent localised solution.

Next we analyse the corresponding self-consistency of the delocalised phase. Since $\tilde{\Delta}_{\text{typ}}$ in this phase is finite, the limit $\tilde{\eta} = 0$ can be taken from the outset, and the self-consistent $\tilde{\Delta}_{\text{typ}}$ for the distribution Eq. (28) can be directly computed as

$$\begin{aligned}
\log \tilde{\Delta}_{\text{typ}} &= \int_{-\infty}^\infty d\tilde{\mathcal{E}}_I \, P_{\tilde{\mathcal{E}}}(\tilde{\mathcal{E}}_I) \int_0^\infty d\tilde{\Delta} \; \delta\left(\tilde{\Delta} - \frac{\overline{\Gamma^2}\tilde{\Delta}_{\text{typ}}}{\tilde{\mathcal{E}}_I^2 + \tilde{\Delta}_{\text{typ}}^2}\right) \log \tilde{\Delta} \\
&= \frac{1}{\sqrt{2\pi\mu_{\tilde{\mathcal{E}}}^2}} \int_{-\infty}^\infty d\tilde{\mathcal{E}}_I \, \exp\left(-\frac{\tilde{\mathcal{E}}_I^2}{2\mu_{\tilde{\mathcal{E}}}^2}\right) \log\left[\frac{\overline{\Gamma^2}\tilde{\Delta}_{\text{typ}}}{\tilde{\mathcal{E}}_I^2 + \tilde{\Delta}_{\text{typ}}^2}\right].
\end{aligned} \tag{36}$$

The integral here can be reorganised in the form

$$\log \tilde{\Delta}_{\text{typ}} = \log\left(\frac{2e^\gamma \overline{\Gamma^2}}{\mu_{\tilde{\mathcal{E}}}^2} \tilde{\Delta}_{\text{typ}}\right) - \frac{1}{\sqrt{2\pi\mu_{\tilde{\mathcal{E}}}^2}} \int_{-\infty}^\infty d\tilde{\mathcal{E}}_I \, \exp\left(-\frac{\tilde{\mathcal{E}}_I^2}{2\mu_{\tilde{\mathcal{E}}}^2}\right) \log\left(1 + \frac{\tilde{\Delta}_{\text{typ}}^2}{\tilde{\mathcal{E}}_I^2}\right). \tag{37}$$

Since $\tilde{\Delta}_{\text{typ}}$ vanishes as the transition is approached from the delocalised side, in the vicinity of the critical point only the low-$\tilde{\Delta}_{\text{typ}}$ behaviour of the integral in Eq. (37) is required. From it, the self-consistency condition is obtained as

$$\tilde{\Delta}_{\text{typ}} \stackrel{\tilde{\Delta}_{\text{typ}} \ll 1}{=} \frac{2e^\gamma \overline{\Gamma^2}}{\mu_{\tilde{\mathcal{E}}}^2} \tilde{\Delta}_{\text{typ}} \left(1 - \frac{\sqrt{2\pi}}{\mu_{\tilde{\mathcal{E}}}} \tilde{\Delta}_{\text{typ}} + \frac{[1+\pi]}{\mu_{\tilde{\mathcal{E}}}^2} \tilde{\Delta}_{\text{typ}}^2 + \mathcal{O}(\tilde{\Delta}_{\text{typ}}^3)\right). \tag{38}$$

Since $\tilde{\Delta}_{\text{typ}}$ is necessarily non-negative, Eq. (38) has a non-trivial solution only for

$$\Lambda = \frac{2e^{\gamma}\overline{\Gamma^2}}{\mu_{\tilde{\mathcal{E}}}^2} \geq 1, \tag{39}$$

with the equality denoting the boundary in parameter space beyond which the delocalised phase fails to exist self-consistently. In addition, as $\Lambda \to 1+$ and the transition is approached, $\tilde{\Delta}_{\text{typ}} \propto [\Lambda - 1]$ is seen to vanish continuously, with a critical exponent of unity.

It is important to note that self-consistency for the many-body localised and delocalised phases, calculated separately, breaks down at precisely the same set of points as shown in Eqs. (35) and (39). The phase boundary between the two phases is thus given by

$$\Lambda = \frac{2e^{\gamma}\overline{\Gamma^2}}{\mu_{\tilde{\mathcal{E}}}^2} = 1, \tag{40}$$

with $\Lambda < 1$ indicating a many-body localised phase and $\Lambda > 1$ a delocalised phase.

## 4 Phase diagram from mean-field theory

Armed with the criterion for the many-body localisation transition from the mean-field theory, Eq. (40), we now derive the phase-diagram of the model in the parameter space spanned by $\alpha$, $\beta$, and $W$. In particular, we will obtain the critical disorder strength, $W_c$, as a function of the power-law exponents $\alpha$ and $\beta$.

From Eq. (40), it is clear that the critical boundary is governed by the interplay between $\overline{\Gamma^2}$ and $\mu_{\tilde{\mathcal{E}}}^2$. Inspecting the expressions for them, Eqs. (16) and (26) respectively, clearly shows that the lines $\alpha = 1/2$ and $\beta = 1/2$ are natural boundaries in the $\alpha$-$\beta$ plane. As such, the regions separated by them warrant separate analyses.

In the following, we analyse these regions systematically:

I. $\boldsymbol{\beta \leq 1/2}$ and $\boldsymbol{\beta < \alpha}$. This can be separated into three sub-regions:

- $\boldsymbol{\alpha > 1/2}$. In this region, $\overline{\Gamma^2} \sim N/\mu_E^2$. By contrast, $\mu_{\tilde{\mathcal{E}}}^2 \sim N \log N/\mu_E^2$ for $\beta = 1/2$ and $\sim N^{2-2\beta}/\mu_E^2$ for $\beta < 1/2$; whence $\Lambda \sim (\log N)^{-1}$ and $N^{2\beta-1}$ in the two cases respectively. Hence, in the thermodynamic limit $N \to \infty$, $\Lambda$ vanishes for both $\beta = 1/2$ and $\beta < 1/2$. The system is thus always many-body localised in this domain, and no transition exists.

- $\boldsymbol{\alpha = 1/2}$. In this case, $\overline{\Gamma^2} \sim N \log N/\mu_E^2$. Since $\beta < \alpha$, we thus consider $\beta < 1/2$. So in this sub-region, $\Lambda \sim N^{2\beta-1} \log N \to 0$ as $N \to \infty$. Hence, as for the previous sub-region, the system is always many-body localised.

- $\boldsymbol{\alpha < 1/2}$. Here, $\overline{\Gamma^2} \sim N^{2-2\alpha}/\mu_E^2$. So for $\beta < 1/2$, $\Lambda \sim N^{2(\beta-\alpha)} \to 0$ as $N \to \infty$ owing to $\beta < \alpha$. Here too the system is therefore always localised.

This analysis shows that throughout the region defined by $\beta \leq 1/2$ and $\beta < \alpha$ (shown in dark blue in Fig. 2(a)), the system is always many-body localised in the thermodynamic limit.

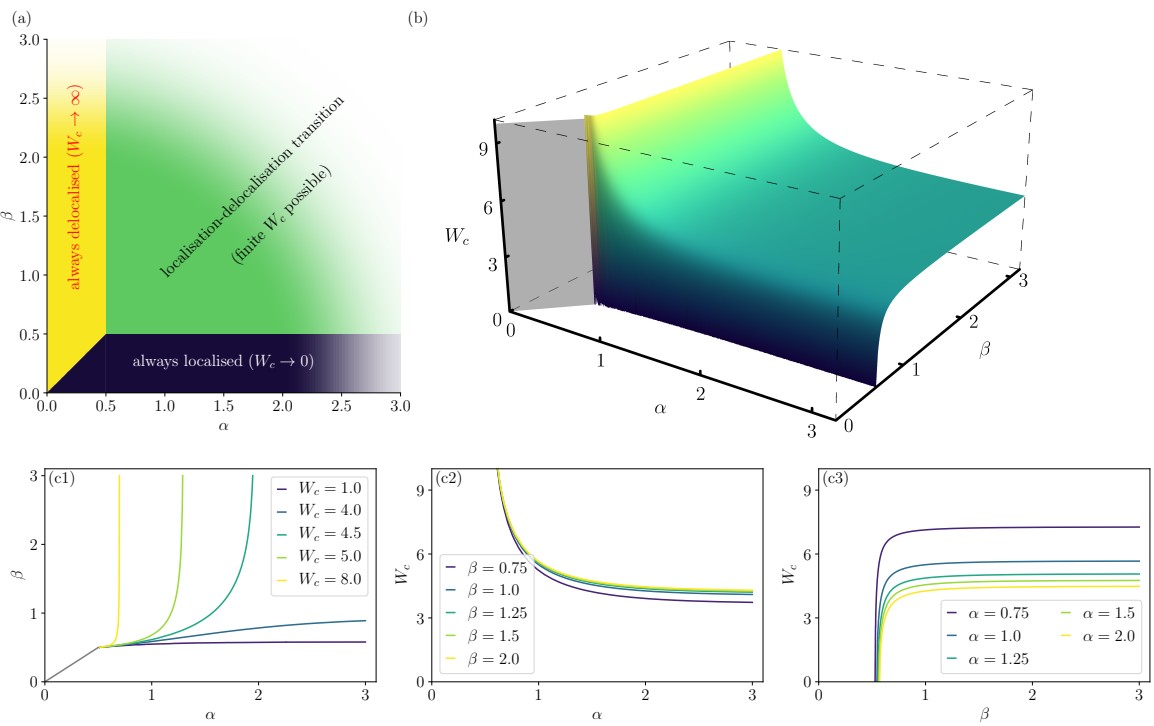

Figure 2: **Mean-field phase diagram:** (a) Schematic of the phase diagram in the $\alpha$-$\beta$ plane obtained from the scaling of $\overline{\Gamma^2}$ and $\mu_{\tilde{\mathcal{E}}}^2$ with $N$ in the thermodynamic limit. Within the yellow [dark blue] region the system is always delocalised [localised], as the critical disorder diverges [vanishes] in the thermodynamic limit. In the green region, both $\overline{\Gamma^2}$ and $\mu_{\tilde{\mathcal{E}}}^2$ scale in the same way with $N$, and there is thus a finite critical disorder. Note that the boundaries between the green and the yellow [dark blue] regions corresponding to $\alpha = 1/2$ [$\beta = 1/2$] are not phase boundaries (critical lines), but simply represent no-go regions for the localised [delocalised] phases. The actual critical lines on the $\alpha$-$\beta$ plane (which depend on the disorder strength) are shown in (c1). (b) The critical disorder surface, $W_c(\alpha, \beta)$, obtained from the mean-field treatment, Eq. (42), shown as a function of $\alpha$ and $\beta$, with $J_z = J$ ($\equiv 1$). The region above and below the surface corresponds respectively to localised and delocalised phases. (c1)-(c3) Sections of this surface are shown in their complementary planes of constant $W_c$, $\beta$ and $\alpha$. The constant $W_c$ contours (c1) in the $\alpha$-$\beta$ plane shift towards low $\alpha$ and high $\beta$ regions as $W_c$ is increased (the localised phase lies below and to the right of the lines shown). The constant $\beta$ contours (c2), and constant $\alpha$ contours (c3), clearly show respectively that $W_c$ increases with decreasing $\alpha$, and decreases with decreasing $\beta$.

II. **$\alpha \leq 1/2$ and $\beta > \alpha$.** In this region, $\overline{\Gamma^2} \sim N \log N/\mu_E^2$ for $\alpha = 1/2$ and $\sim N^{2-2\alpha}/\mu_E^2$ for $\alpha < 1/2$. As in the previous case, we analyse the region by splitting it into three sub-regions:

- **$\beta > 1/2$.** In this case $\mu_{\tilde{\mathcal{E}}}^2 \sim N/\mu_E^2$. Hence, $\Lambda \sim \log N$ and $N^{1-2\alpha}$ for $\alpha = 1/2$ and $\alpha < 1/2$ respectively. In either case, $\Lambda \to \infty$ as $N \to \infty$. Consequently the system is delocalised for any finite value of $W$.

- **$\beta = 1/2$.** On this line, $\mu_{\tilde{\mathcal{E}}}^2 \sim N \log N/\mu_E^2$. Since $\beta > \alpha$, only $\alpha < 1/2$ is relevant here. Hence $\Lambda \sim N^{1-2\alpha}/\log N$, which diverges in the thermodynamic limit, showing that here too the system is always delocalised.

- **$\beta < 1/2$.** Here, $\mu_{\tilde{\mathcal{E}}}^2 \sim N^{2-2\beta}/\mu_E^2$ and hence $\Lambda \sim N^{2(\beta-\alpha)}$. Since $\beta > \alpha$, $\Lambda$ again diverges as $N \to \infty$, making localisation impossible.

Hence, throughout the region defined by $\alpha \leq 1/2$ and $\beta > \alpha$ (shown in yellow in Fig. 2(a)), the system is always delocalised in the thermodynamic limit.

III. **$\alpha > 1/2$ and $\beta > 1/2$.** In this region of the $\alpha$-$\beta$ plane, shown in green in Fig. 2(a), both $\overline{\Gamma^2}$ and $\mu_{\tilde{\mathcal{E}}}^2$ scale as $N/\mu_E^2$. Consequently $\Lambda$ is finite in the thermodynamic limit, and the interplay between $J$, $J_z$, and $W$ can lead to a phase transition at a finite critical $W_c$, which naturally depends on $\alpha$ and $\beta$. In this regime $\mu_{\mathcal{E}}^2 = \mu_{\text{int}}^2 + \mu_{\text{dis}}^2$, where $\mu_{\text{dis}} \propto W$ is the contribution due to the external disorder strength $W$ arising from the disordered fields, and $\mu_{\text{int}} \propto J_z$ is the contribution to $\mu_{\mathcal{E}}$ from the interactions, reflecting the configurational disorder in the Fock-space basis states. From Eq. (26), one can read off $\mu_{\text{int}}^2 = J_z^2 \zeta(2\beta)N$ and $\mu_{\text{dis}}^2 = W^2 N/3$, such that $\mu_{\tilde{\mathcal{E}}}^2 = [J_z^2 \zeta(2\beta) + W^2/3]N/\mu_E^2$. Additionally, from Eq. (16), in this region $\overline{\Gamma^2} = \frac{2J^2}{\mu_E^2} N \zeta(2\alpha)$. Hence from Eq. (40),

$$\Lambda = 4e^{\gamma} \frac{J^2 \zeta(2\alpha)}{J_z^2 \zeta(2\beta) + W^2/3}, \tag{41}$$

which when set to unity yields an expression for the critical disorder,

$$W_c = \sqrt{3}\sqrt{4e^{\gamma}J^2\zeta(2\alpha) - J_z^2\zeta(2\beta)}. \tag{42}$$

The Riemann zeta function $\zeta(s)$ diverges as $s \to 1+$, but decreases rapidly and monotonically with increasing $s$ towards its asymptotic limit $\zeta(\infty) = 1$ (such that $\zeta(s)$ is within a few percent of unity for $s \gtrsim 4$).

The resultant critical disorder surface $W_c(\alpha, \beta)$ represented by Eq. (42) is shown in Fig. 2(b) for the case $J_z = J$, while sections of it in the complementary planes of constant $W_c$, $\alpha$ and $\beta$ are given in Fig. 2(c1-3). Note that at a fixed $\alpha$, decreasing $\beta$ decreases $W_c$; in other words, increasing the range of the interaction in the longitudinal direction drives the system more towards a many-body localised phase. The critical disorder strength eventually falls to zero (see also Fig. 2(c3)) at a value $\beta_c \geq 1/2$ given by $\zeta(2\beta_c) = 4e^{\gamma}J^2\zeta(2\alpha)/J_z^2$. On the other hand, decreasing $\alpha$ at a fixed $\beta$ acts to delocalise the system, as indicated by a growing $W_c$. As $\alpha$ decreases, $W_c$ rises towards infinity (see also Fig. 2(c2)), and for $\alpha < \alpha_c$ the system is inexorably delocalised, with $\alpha_c \geq 1/2$ given by $\zeta(2\alpha_c) = J_z^2\zeta(2\beta)/(4e^{\gamma}J^2)$.

In summary, for $\alpha, \beta > 1/2$ a many-body localisation transition is possible at a finite critical disorder strength given by Eq. (42). Increasing the range of the interactions in

the transverse direction favours delocalisation while, in marked contrast, increasing the range of the longitudinal interactions favours localisation.

For completeness, we reiterate that the lines $\alpha = 1/2$ and $\beta = 1/2$ are not phase boundaries, but simply define regions (Fig. 2(a)) where localisation or delocalisation is forbidden owing to the scaling arguments discussed in points I and II above. The actual mean-field phase boundaries, given by Eq. (**??**), can lie well away from these lines, as also illustrated in Fig. 2(c1).

The only part of the $(\alpha, \beta)$-plane not included in the above analysis is the line segment $\alpha = \beta < 1/2$. Here the disorder strength $W$ is irrelevant as $\mu_{\mathcal{E}}^2$ is completely dominated by $\mu_{\text{int}}^2$, but both $\overline{\Gamma^2}$ and $\mu_{\tilde{\mathcal{E}}}^2$ scale as $N^{2(1-\alpha)}/\mu_E^2$ so a localisation transition driven by the ratio of $J/J_z$ can thus in principle lie on this line. We do not however pursue it further here, both because our primary interest is in possible transitions driven by the disorder strength $W$, and because this line segment is likely to be rather delicate, surrounded as it is on either side (Fig. 2(a)) by phases which are exclusively either delocalised or localised.

## 5   Numerical results

While the mean-field treatment allows us to derive analytically a phase diagram for the model in the thermodynamic limit, it is of course approximate. It is thus important to compare the mean-field phase diagram to that obtained from standard numerical diagnostics, which are free from the approximations underlying the mean-field theory. In this section we obtain representative sections of the phase diagram numerically, using three ubiquitous and complementary diagnostics: the statistics of level-spacing ratios, and participation entropies and entanglement entropies of eigenstates. It should be kept in mind that the largest system size ($N = 18$ spins) accessed with our exact diagonalisation calculations is naturally quite far from the thermodynamic limit, and possibly also not in the scaling regime. Finite-size effects are in fact significant already in the short-ranged MBL problem, and in the context of long-ranged interactions it is natural to expect them to be worse. The numerical results presented here should not therefore be viewed as quantitatively definitive. Nevertheless, we will show that finite-size scaling analyses of the above range of diagnostics demonstrate clearly that decreasing $\beta$ at a fixed $\alpha$ decreases the critical disorder strength $W_c$, while decreasing $\alpha$ at a fixed $\beta$ enhances $W_c$; consistent with the mean-field phase diagram obtained in Sec. 4.

We first describe briefly the three numerical diagnostics, and their expected behaviour in the two phases. The level spacing ratio $r_\alpha$ is defined as [6, 45]

$$r_\alpha = \frac{\min(s_\alpha, s_{\alpha-1})}{\max(s_\alpha, s_{\alpha-1})} \qquad : \ s_\alpha = E_\alpha - E_{\alpha-1}, \tag{43}$$

where $E_{\alpha-1}, E_\alpha$ are consecutive eigenvalues of the Hamiltonian Eq. (1). Ergodic systems are well described by random matrix ensembles, with a Wigner-Dyson distribution for $r$ depending on the symmetries (in our case, the Gaussian Orthogonal Ensemble (GOE)). In a non-ergodic localised phase by contrast the energy levels are uncorrelated, with absence of level repulsion leading to a Poisson distribution. For the former the mean $\langle r \rangle \simeq 0.53$, and for the latter $\langle r \rangle \simeq 0.38$ [45]. For a model hosting a many-body localisation transition as a function of disorder, $\langle r \rangle$ for a finite system crosses over from the GOE value to the Poisson value, with

the data for various system sizes showing crossings as $N$ is varied. The critical disorder is estimated by collapsing the $\langle r \rangle$ for various system sizes onto a common scaling function of the form $g_r[(W - W_c)N^{1/\nu}]$ (with $\nu$ the correlation length exponent).

In addition to spectral properties, many-body localisation also manifests itself in real space via a transition of the bipartite entanglement entropy measured on an eigenstate, from an area law in the localised phase to a volume law in the delocalised phase [16, 17, 46, 47]. For an eigenstate $|\psi\rangle$, the entanglement entropy of the left-half of the chain (L) with the right-half (R) of the chain is given by

$$S^E = -\mathrm{Tr}[\rho_L \log \rho_L], \tag{44}$$

where $\rho_L = \mathrm{Tr}_R \rho$ and $\rho = |\psi\rangle \langle\psi|$ (with $\mathrm{Tr}_R$ representing the partial trace over the right-half of the system). In the localised phase, $S^E \sim N^0$, while in the delocalised phase $S^E \sim N$. Deep in the delocalised phase in particular, one expects the entanglement to be close to that of a random state in Hilbert space, i.e. $S^E = N \log 2 - 1/2$ [48]; adding that for models with conserved quantities, such as $M_z$ in our case, the conservation leads to a slight deficit from the maximal entanglement value (see Ref. [49] for details), which is evident in the results shown below. For a finite system, the critical disorder can be obtained by noting that $S^E/N$ plotted against $W$ also shows a crossing for various $N$, whence the data can be collapsed onto a common scaling form $g_s[(W - W_c)N^{1/\nu}]$. In addition, the fluctuations of $S^E$ over disorder realisations, as measured by their standard deviation, $\sigma^E$, also show a peak at the localisation transition [16, 17].

Finally, since many-body localisation is a Fock-space phenomenon, its signatures are also revealed by participation entropies of the eigenstates $|\psi\rangle$ [17, 39, 40], defined by

$$S_q^P(|\psi\rangle) = \frac{1}{1-q} \log \sum_I |\langle\psi|I\rangle|^{2q}, \quad S_1^P(|\psi\rangle) = -\sum_I |\langle\psi|I\rangle|^2 \log |\langle\psi|I\rangle|^2; \tag{45}$$

and in particular the first participation entropy $S_1^P$ on which we focus. Similarly to Ref. [17] we analyse the data by fitting it to the form

$$S_1^P = a_1 S_0^P + l_1 \log S_0^P \qquad : \ S_0^P = \log N_{\mathcal{H}} \tag{46}$$

where $a_1 \simeq 1$ in the delocalised phase whereas $a_1 < 1$ in the localised phase.

The above diagnostics are calculated via exact diagonalisation for systems with up to 18 spins. To access band centre states appropriately, we consider only a few tens of eigenstates with their energies close to $\mathrm{Tr}'[\mathcal{H}]$. The case $J = J_z \ (\equiv 1)$ is considered throughout, with statistical errors determined by the standard bootstrap method with 500 resamplings.

We first discuss results in the $(\alpha, W)$-plane, for a fixed value $\beta = 10$. A relatively large $\beta$ is taken so that the longitudinal interactions do not have a particularly long-range, and we can effectively distil out the interplay of $\alpha$ and $W$. The results are shown and described in Fig. 3. Since the critical disorder grows with decreasing $\alpha$, it is more convenient to present the data as a function of inverse disorder $1/W$.

Representative results for $\langle r \rangle$, $S^E$, and $\sigma^E$ versus $1/W$ are shown in panels (a) and (b), for two values of $\alpha \ (= 4$ and $0.1)$, and for system sizes ranging from $N = 8 - 18$. For $\alpha = 4$, which the mean-field theory suggests is connected adiabatically to the $\alpha \to \infty$ (short-ranged) limit, there is a clear crossing of the data for various system sizes in $\langle r \rangle$ as well as in $S^E$, indicating the occurrence of a transition. By contrast, no such crossing appears in the $\alpha = 0.1$ case, and the trend with system size suggests that the system is delocalised at any finite value of $W$ in the thermodynamic limit. This is consistent with the prediction of the mean-field theory.

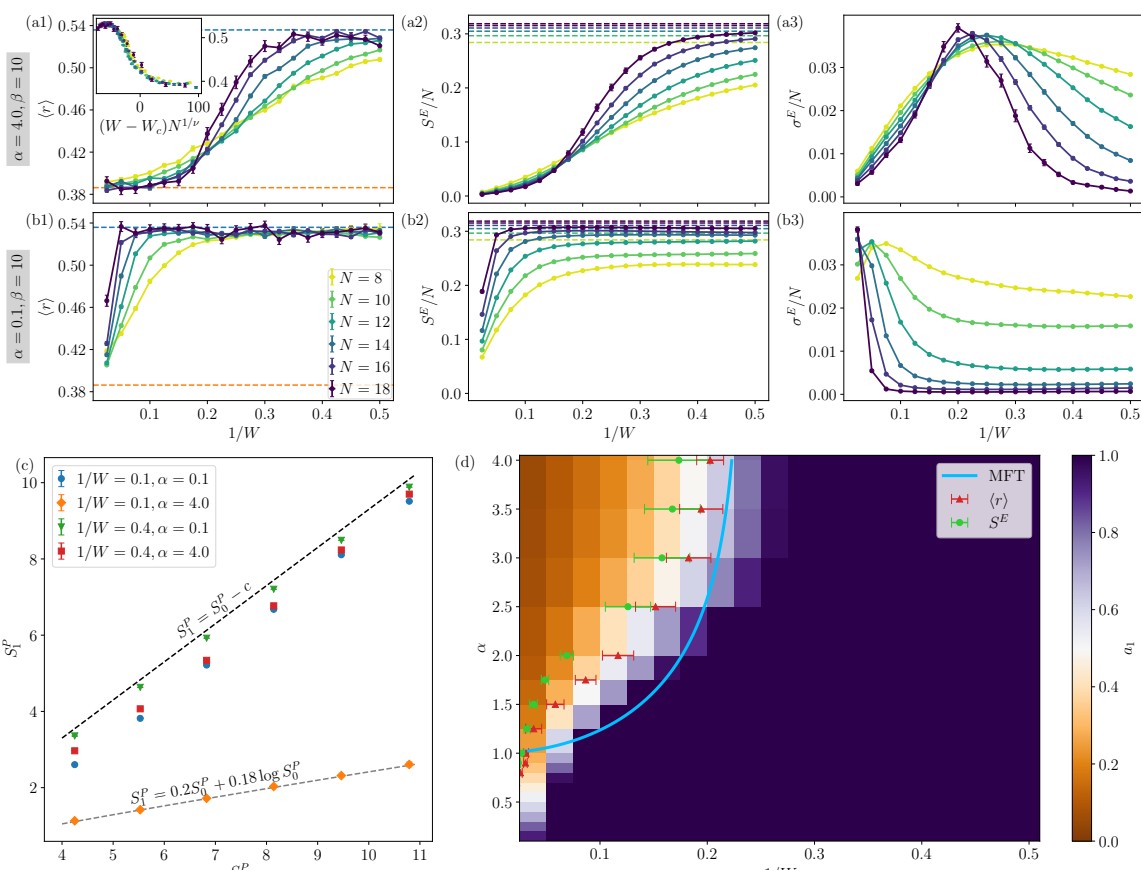

Figure 3: **Numerical phase diagram in a constant-$\beta$ plane:** Top two rows show data for the mean level-spacing ratios $\langle r \rangle$, the mean entanglement entropy $S^E$, and the fluctuations in the entanglement entropy $\sigma^E$, as a function of the inverse disorder strength, $1/W$, for a fixed value of $\beta = 10$ and two values of $\alpha = 4$ (top panels (a1)-(a3)) and $\alpha = 0.1$ (middle panels (b1)-(b3)). Data are shown for $N = 8 - 18$ spins. For $\alpha = 0.1$, $\langle r \rangle$ stays pinned to the GOE value even for large $W$ and there is no visible crossing of the data for various $N$, indicating the absence of the MBL phase. For $\alpha = 4$ on the other hand, there is a clear crossing of the data at finite $W$ suggesting a transition, with $\langle r \rangle$ going to the GOE and Poisson values (blue and orange dashed lines respectively) at weak and strong disorder; the inset to panel (a1) gives the scaling function $g_r[(W - W_c)N^{1/\nu}]$, which shows good scaling collapse. The half-chain entanglement entropy $S^E/N$ also shows the same behaviour: for $\alpha = 0.1$ it remains very close to the Page corrected volume-law value $S^E/N = \log 2 - 1/(2N)$ [48] (dashed lines) even for large $W$, whereas for $\alpha = 4$ there is a clear crossing of the data suggesting a transition. Fluctuations in entanglement entropy are likewise consistent, showing for $\alpha = 0.1$ that with increasing $N$ its peak shifts to progressively higher values of $W$, suggesting a delocalised phase throughout in the thermodynamic limit; whereas the peak for $\alpha = 4$ lies quite close to the critical $W_c$ predicted by the $\langle r \rangle$ and $S^E$ data. Panel (c) shows the scaling of the first participation entropy $S_1^P$ with the logarithm of the Fock-space dimension, $S_0^P = \log N_{\mathcal{H}}$, for two representative values of $W$ and two values of $\alpha$. While for $\alpha = 0.1$ both values of $W$ show $a_1 \simeq 1$, for $\alpha = 4$, $a_1$ changes from $\simeq 1$ at $W = 0.25$ to $\simeq 0.2$ at $W = 10$, indicating the occurrence of a transition. The $a_1$ values in the $(\alpha, 1/W)$-plane are shown a colour-map in panel (d), clearly showing $W_c$ moves to higher values as $\alpha$ is decreased. The $W_c$ values extracted from finite-size scaling analyses of $\langle r \rangle$ and $S^E$ are also shown, and are concomitant with the phase boundary predicted from participation entropies. These phase boundaries are remarkably consistent overall with the prediction from the mean-field theory, which is shown by the light blue line in panel (d).

Further, the coefficient $a_1$ defined in Eq. (46) can be computed by fitting the participation entropy data to the form Eq. (46), as shown in Fig. 3(c). The $a_1$ value thus extracted for a set of points in the $(\alpha, 1/W)$-plane can be plotted as a colour-map as in Fig. 3(d), which clearly shows the phase boundary between the many-body localised and delocalised phases. Finite-size scaling analyses of $\langle r \rangle$ and $S^E/N$ have been performed; an example of the scaling function $g_r$ is given in the inset to Fig. 3(a1) (for $\alpha = 4$), and shows good scaling collapse. These analyses of $\langle r \rangle$ and $S^E/N$ also yield critical $W_c$ values consistent with, and quite close to, the phase boundary resulting from $a_1$, as likewise shown in Fig. 3(d). We add that, where we obtain a transition via exact diagonalisation, an exponent $\nu \approx 1$ is found, which violates the Harris/CCFS bounds [50,51] (requiring $\nu \geq 2/d$ with $d$ the space dimension). Reflecting finite-size effects, this is also as found in other exact diagonalisation studies [17,52].

While the quantitative accuracy of the numerically-determined phase diagram in Fig. 3(d) could be questioned owing to finite-size limitations, it is nevertheless seen to be remarkably consistent overall with the prediction from mean-field theory, the corresponding phase boundary for which is also shown in the figure (light blue line). The qualitative mean-field prediction that the critical disorder increases with decreasing $\alpha$ is entirely clear in the numerical data; indeed it is also remarkable to note from Fig. 3(d) that the critical $(1/W_c) \to 0$ in the vicinity of $\alpha = 1/2$.

We turn now to the complementary case of a fixed value $\alpha = 10$ for the transverse interaction exponent, and results in the $(\beta, W)$-plane. This is a more difficult case to handle numerically, because the critical disorder decreases from the short-ranged value as $\beta$ is decreased. For small $\beta$, where the mean-field theory predicts localisation at any disorder strength, the system sizes accessible to exact diagonalisation could well be too small to show localisation for small values of $W$, since the interaction range grows with decreasing $\beta$. This leads to an apparent qualitative discrepancy between the numerical and mean-field results, to which we return shortly; but first we describe the results shown in Fig. 4 as they are.

Panels (a) and (b) of Fig. 4 show results for $\beta = 2.8$ and $\beta = 0.1$. In both cases, there appears to be crossing in the data for various system sizes in both $\langle r \rangle$ and $S^E/N$, suggesting a finite $W_c$. However, comparison between panels (a) and (b) shows that the $W_c$ decreases with decreasing $\beta$, and this is so far consistent with the mean-field theory. Similar to the analysis shown in Fig. 3(c), the coefficients $a_1$ can be extracted on a grid of points spanning the $(\beta, W)$-plane. This is shown as a colour-map in Fig. 4(d), with the mean-field phase boundary also indicated (light blue line). The numerical phase boundary predicted by the $a_1$ values, as well those obtained by the finite-size scaling analyses of $\langle r \rangle$ and $S^E$, also seem consistent with each other. The numerical results are, remarkably, concomitant with the prediction of the mean-field theory that the critical disorder strength $W_c$ decreases with decreasing $\beta$. That consistency, even at this level, is rather reassuring because this result goes against the naive expectation that increasing the range of interactions (decreasing $\beta$) always makes the system more vulnerable to delocalisation.

We now return to the qualitative discrepancy between the numerical and mean-field results. Recall from Sec. 4 that the mean-field theory predicted that for $\beta < 0.5$ and $\alpha > \beta$, a delocalised phase is not possible and the system is many-body localised throughout. Yet this is not captured by the numerical results, which show a finite $W_c$ for values of $\beta$ much below $1/2$. We now argue, however, that this is likely to be a finite-size effect. Note that in the

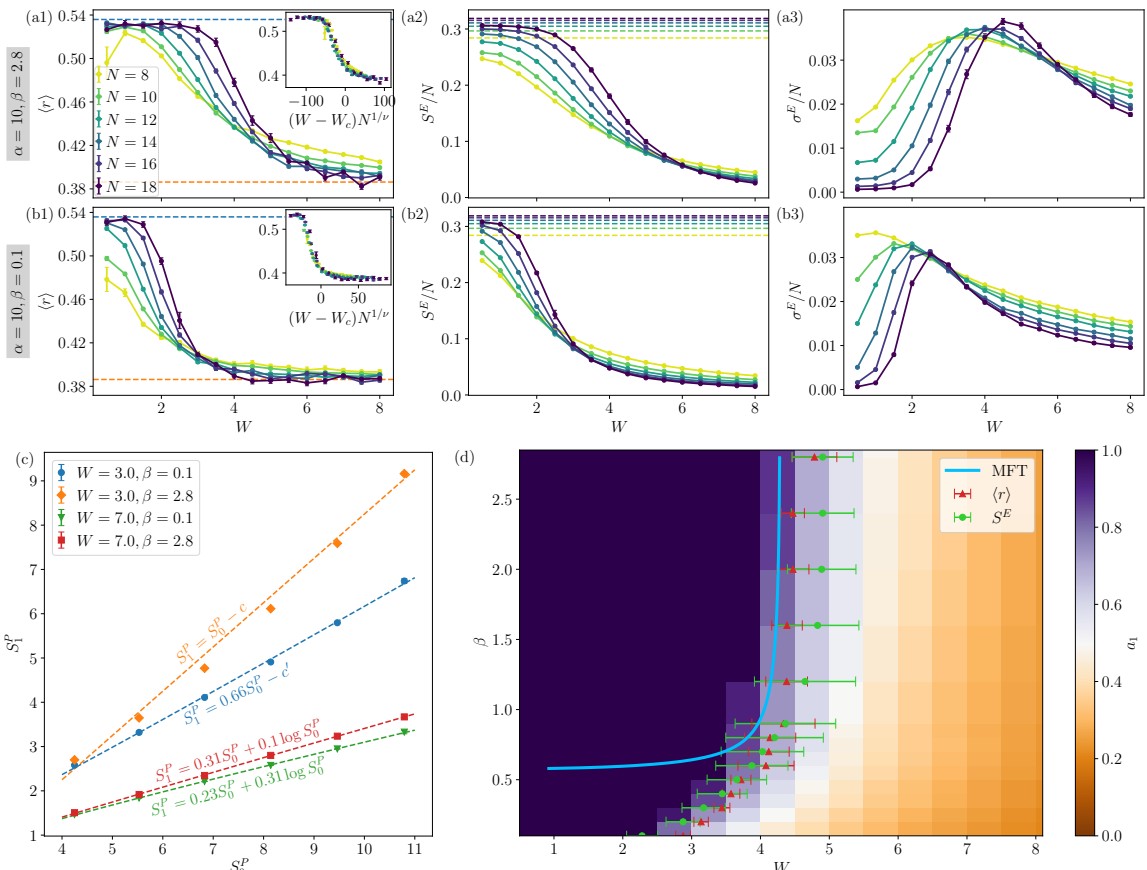

Figure 4: **Numerical phase diagram in a constant-$\alpha$ plane:** Figure is analogous to Fig. 3, but with a constant $\alpha = 10$. Top two rows show data as a function of disorder strength, $W$, for two values of $\beta = 2.8$ (panels (a1)-(a3)) and $\beta = 0.1$ (panels (b1)-(b3)). In the $\langle r \rangle$ data there is an apparent crossing for both values of $\beta$, but $W_c$ is clearly smaller for $\beta = 0.1$ than $\beta = 2.8$. Data for $S^E/N$ and $\sigma^E/N$ convey the same message, again showing that $W_c$ decreases for smaller $\beta$. Panel (c) shows the the first participation entropy $S_1^P$ vs $S_0^P = \log N_{\mathcal{H}}$, for two representative values of $W$ and two values of $\beta$. For large enough disorder, e.g. $W = 7$ in the figure, $a_1 < 1$ for both values of $\beta$, indicating a many-body localised phase. However for $W = 3$, $a_1 \simeq 1$ for $\beta = 2.8$, indicating that the system has transited to a delocalised phase,while for $\beta = 0.1$, $a_1$ continues to be $< 1$, the system thus remaining localised; and again showing that $W_c$ decreases with decreasing $\beta$. The phase diagram in the $(\beta, W)$-plane for $\alpha = 10$ is shown in panel (d). The $a_1$ values are shown as a colour-map, and the $W_c$ values extracted from the finite-size scaling analyses of $\langle r \rangle$ and $S^E$ are indicated; the prediction from mean-field theory is shown by the light blue line. All clearly show that the critical $W_c$ decreases with decreasing $\beta$.

$\beta \to 0$ limit the longitudinal term in the Hamiltonian Eq. (1) can be written as

$$\lim_{\beta \to 0} J_z \sum_{i>j} \frac{1}{(i-j)^\beta} \sigma_i^z \sigma_j^z = \frac{J_z}{2}\left[\left(\sum_i \sigma_i^z\right)^2 - N\right] = -N\frac{J_z}{2}, \tag{47}$$

where the last equality reflects conservation of total magnetisation $M_z$ and that we work in the $M_z = 0$ sector. More importantly, in this limit the longitudinal interaction term is a constant, and hence completely drops out (modulo a constant shift). Next, note that the mean-field theory suggests that the behaviour arising on decreasing $\beta$ for some fixed $\alpha$ ($> 1/2$), is adiabatically connected to that for $\alpha \to \infty$. In the latter limit the transverse interaction is purely short-ranged, so $\mathcal{H}$ in this limit becomes

$$\begin{aligned}
\lim_{\beta \to 0}\lim_{\alpha \to \infty} \mathcal{H} &= \sum_i [J(\sigma_i^x \sigma_{i+1}^x + \sigma_i^y \sigma_{i+1}^y) + h_i \sigma_i^z] + \text{constant}_1 \\
&= \sum_i 2[J(c_i^\dagger c_{i+1} + \text{h.c.}) + h_i c_i^\dagger c_i] + \text{constant}_2
\end{aligned} \tag{48}$$

where the standard Jordan-Wigner transformation is used in the second line. The resulting fermionic model is simply the Anderson model in one-dimension, which is well known to be localised for infinitesimally weak disorder. Hence for $\alpha \to \infty$, the $\beta = 0$ line is completely localised. One can argue that for $\beta = 0^+$ and $\alpha \gg 1$ ($\alpha = 10$ is considered in Fig. 4) the system stays localised, suggesting that the finite $W_c$ at $\beta = 0^+$ in Fig. 4(d) is a finite-size effect. Further discussion of this point, in the context of spinless fermion models, is given in Sec. 6 below.

# 6   Discussion

In summary, the problem of many-body localisation in a long-ranged interacting quantum spin chain has been considered analytically, using a self-consistent mean-field treatment of the self-energy associated with the local Fock-space propagator, and our essential results have been confirmed by numerics obtained from exact diagonalisation.

In particular, we studied an XXZ chain with disordered fields coupling locally and independently to the longitudinal spin component, and with power-law decaying interactions characterised by exponents $\beta$ and $\alpha$, respectively, for longitudinal and transverse spin-spin interactions. A central result of the work has been a derivation of the localisation phase diagram of the model in the parameter space spanned by the power-law decay exponents, and the disorder strength. Increasing the range of the transverse interaction was found to make the system more susceptible to delocalisation, with the critical disorder increasing upon decreasing $\alpha$. By contrast, increasing the range of the longitudinal interaction provides the system with a rigidity against spin flips, which cooperates with the external disorder and makes localisation increasingly favourable. This is reflected in the fact that the critical disorder decreases with decreasing $\beta$. In fact, the mean-field theory goes so far as to predict that for $\beta < 1/2$ and $\beta < \alpha$, the system is always many-body localised even in the absence of external disorder, much like an interaction-induced localised phase. On the contrary, for $\alpha < 1/2$ and $\alpha < \beta$, the mean-field theory concludes that localisation is impossible at any finite disorder strength.

Our results call for discussion of two important and related questions. First, what do they imply for a disordered model of spinless fermions with long-ranged hoppings and long-ranged density-density interactions? Unlike the nearest-neighbour models, the fermionic model is not trivially equivalent to the spin-1/2 chain, due to the presence of non-local Jordan-Wigner strings. Second, if longer-ranged fermionic density-density interactions are correspondingly found within mean-field theory to enhance localisation, can a physical rationale be given for such behaviour, given that it goes against the common lore that long-ranged interactions generally act to suppress localisation?

To put the question in context, the problem of non-interacting fermions with random power-law hoppings has a long history [53–60], with applications in dipolar systems, Anderson transitions, and quantum Hall plateau transitions, and has generated exotic phenomena such as power-law localised and multifractal wavefunctions. A different phenomenology arises for non-interacting fermionic models with onsite disorder but *non-random* power-law hoppings, which have also attracted considerable attention [61–68]. As discussed below, it is this case that is relevant to our considerations.

To make a connection to our mean-field theory results, we note that they are in fact insensitive to the presence of long-ranged Jordan Wigner strings. Consider for concreteness

$$H = \sum_{i>j} \left[ \frac{t}{r_{ij}^\alpha} \left( c_i^\dagger c_j + \text{h.c.} \right) + \frac{V}{r_{ij}^\beta} \hat{n}_i \hat{n}_j \right] + \sum_i \epsilon_i \hat{n}_i, \tag{49}$$

where $\hat{n}_i = c_i^\dagger c_i$ and $\epsilon_i \in [-W_f, W_f]$ is the disordered onsite potential. The mean-field localisation criterion (embodied in $\Lambda = 1$, Eq. (40)) depends in essence on the ratio of the average weighted connectivities on the Fock-space graph, and the effective disorder in the Fock space as measured by the width of the distribution of Fock-space site energies. The latter is identical for the spin chain and fermionic chain, with the identification $J_z = V/4$ and $W = W_f/2$. The average weighted connectivities count the number of ways of flipping two antiparallel spins at a distance $r$, and sum it with weight $(2J)^2/r^{2\alpha}$. In the fermionic model, the mean-field treatment would count the number of ways of having an occupied and unoccupied site at separation $r$ and sum it with weight $t^2/r^{2\alpha}$, hence yielding the same result as for the spin chain, with the identification $J = t/2$. The mean-field treatment would thus predict the fermionic chain always to be many-body localised for $\beta < 1/2$ and $\beta < \alpha$.

The results at small but finite $\beta$ warrant further elaboration. First, consider the limit of $\beta \to 0$ where, using the fact that total particle number $\sum_i \hat{n}_i$ is conserved, the interaction term can be expressed as

$$\lim_{\beta \to 0} \sum_{i>j} \frac{V}{r_{ij}^\beta} \hat{n}_i \hat{n}_j = \frac{V}{2} \left[ \frac{N^2}{4} - \frac{N}{2} \right] \tag{50}$$

(considering for specificity the case of half-filling, the counterpart of $M_z = 0$). Since this is a constant, it drops out of the Hamiltonian. The model then reduces simply to one of non-interacting fermions with a disordered onsite potential and *non-random* power-law hoppings [61–68]. In such systems, due to a phenomenon termed cooperative shielding [65, 66], Anderson localisation is found to persist for all values of the disorder strength and power-law decay exponent $\alpha$, and for all single-particle states save for a set of measure zero near one edge of the spectrum (which are delocalised for $\alpha < 1$). This implies that generic many-body states, constructed out of Slater determinants of the localised single-particle eigenstates, are

also many-body localised. Hence, on the $\beta = 0$ line, the system is many-body localised for all values of $\alpha$ and $W$ in 1D. Note that this also suggests that the apparent finite $W_c$ for $\beta \to 0^+$ found from numerics (Fig. 4(d)) is indeed a finite-size effect.

Second, consider the case of $\beta \gtrsim 1$. From the reasonably good match between the critical lines obtained from the mean-field treatment and exact diagonalisation, as shown in Fig. 4(d), one can confidently predict that there exists a finite critical disorder strength (and an ensuing many-body localised phase) in this regime. One can also conclude that the critical disorder strength grows with $\beta$ and saturates as $\beta \to \infty$ to its value for the nearest-neighbour XXZ model.

Since there is no evidence of non-monotonicity in the phase diagram, either with $W$ or with $\beta$, the above arguments suggest only two plausible scenarios: (i) the critical disorder vanishes at a finite value of $\beta$, or (ii) it vanishes as $\beta \to 0$. While the mean-field theory predicts the former, determining this precise limiting value of $\beta$ naturally calls for further work. However, what still stands firm is that increasing the range of longitudinal interactions favours localisation and the critical disorder grows with $\beta$.

Whether the aforementioned measure-zero delocalised states at the single-particle spectral edge could conjecturally seed a so-called 'avalanche instability' [69], eventually destroying localisation, is a speculative question which would clearly require a much more refined analysis. In the shorter term, an interesting question for further study is whether dynamical signatures [70, 71] are consonant with the phase diagram derived in this work. As an example, it was recently found that in long-ranged interacting systems in the absence of disorder, the entanglement entropy grows logarithmically in time, much like many-body localised systems [72].

*Note:* During the review process of this paper, another article appeared which reports numerical results qualitatively consistent with those presented here [73].

## Acknowledgements

We are grateful for helpful discussions with Y. Bar Lev, G. De Tomasi, I. M. Khaymovich, H. R. Krishnamurthy, A. Lazarides, D. J. Luitz and S. Welsh. One of us (DEL) expresses his gratitude for support from the Infosys Foundation during his tenure as Infosys Visiting Chair Professor at the Indian Institute of Science, Bangalore, and for the warm hospitality of the IISc Physics Department.

**Funding information**   This work was supported by EPSRC Grant No. EP/N01930X/1.

## A   Derivation of variance of Fock-space site energies

In this appendix, we present details of the derivation of the variance, $\mu_{\mathcal{E}}$, of the Fock-space basis state energies. In particular, we show how the asymptotic forms of the $\Upsilon$s defined in Eq. (24) can be obtained, which ultimately lead to the asymptotic forms of $\overline{\mathcal{E}^2}$ in Eq. (25). Note that in Eq. (24), the summations are over sites with constraints on the terms. The strategy we employ to analyse these summations is to convert them from sums over sites to distances, taking the combinatorial factors into account.

We start with the simplest case, namely, that of $\Upsilon_2$ which is nothing but the sum over all distances, $\ell$, of $\ell^{-2\beta}$ weighted by the number of ways in which two sites in the system can be separated by a distance $\ell$. Hence,

$$\Upsilon_2 = \sum_{\ell=1}^{N-1} \frac{(N-\ell)}{\ell^{2\beta}} = N \sum_{\ell=1}^{N-1} \frac{1}{\ell^{2\beta}} - \sum_{\ell=1}^{N-1} \frac{1}{\ell^{2\beta-1}} \stackrel{N \gg 1}{\approx} \begin{cases} N\zeta(2\beta); & \beta > 1/2 \\ N \log N; & \beta = 1/2 \\ N^{2-2\beta}; & \beta < 1/2 \end{cases}, \tag{51}$$

where the limiting asymptotic forms can be found by replacing the summations with integrations. In fact, for $\beta \geq 1/2$, the summation can be exactly computed in the thermodynamic limit. For $\beta$ strictly greater than $1/2$, the first summation dominates and the result is the Riemann zeta function by its definition. Hence $\Upsilon_2 = N\zeta(2\beta)$ for $\beta > 1/2$. For $\beta = 1/2$, again the first summation dominates and the result is $N \sum_{\ell=1}^{N-1} \ell^{-1}$. Using the property of the Harmonic sum, $\sum_{\ell=1}^{k} \ell^{-1} \stackrel{k\to\infty}{=} \log k$, one arrives at $\Upsilon_2 = N \log N$ for $\beta = 1/2$. For $\beta < 1/2$, the coefficient of $N^{2(1-\beta)}$ can be obtained by evaluating the summations directly (although explicit knowledge of it is not in fact required).

We next consider $\Upsilon_1$, which consists of the terms where there is one common site. Hence, it can be expressed as

$$\Upsilon_1 = \sum_{i\neq j, i\neq l, j\neq l} \frac{1}{|i-j|^\beta |i-l|^\beta} = 2 \sum_{j>i, i\neq l, j\neq l} \frac{1}{|i-j|^\beta |i-l|^\beta}. \tag{52}$$

The last term in the above equation above can be split up into two cases, (i) $l < i$ and (ii) $l > i$, and one can express

$$\Upsilon_1 = 2 \left[ \sum_{j>i, l<i} \frac{1}{|i-j|^\beta |i-l|^\beta} + \sum_{j>i, l>i, l\neq j} \frac{1}{|i-j|^\beta |i-l|^\beta} \right] \tag{53}$$

where the $l \neq j$ constraint is automatically accounted for in the first term. In order to do that for the second term, we let the summation over $l$ run freely and subtract the contribution coming from $l = j$. Hence

$$\Upsilon_1 = 2 \left[ \sum_{j>i, l<i} \frac{1}{|i-j|^\beta |i-l|^\beta} + \sum_{j>i, l>i} \frac{1}{|i-j|^\beta |i-l|^\beta} - \sum_{j>i} \frac{1}{|i-j|^{2\beta}} \right]. \tag{54}$$

In the next step, we convert the summation from sites to distances. Note that for a given $i$, the summation over $j$ constrained to $j > i$ corresponds to summing over distances which lie in the range from $1$ to $N - i$. Similarly, summing over $l$ subject to the constraint $l < i$ is equivalent to summing over distances from $1$ to $i - 1$. Hence, $\Upsilon_2$ can be expressed in terms of summations over distances as

$$\Upsilon_1 = 2 \left[ \sum_{i=2}^{N-1} \sum_{\ell_1=1}^{N-i} \sum_{\ell_2=1}^{i-1} \frac{1}{\ell_1^\beta \ell_2^\beta} + \sum_{i=1}^{N-1} \sum_{\ell_1=1}^{N-i} \sum_{\ell_2=1}^{N-i} \frac{1}{\ell_1^\beta \ell_2^\beta} - \sum_{\ell=1}^{N-1} \frac{N-\ell}{\ell^{2\beta}} \right]. \tag{55}$$

In the limit of $N \gg 1$, the summations are well approximated by integrations over the distances, which yield

$$\Upsilon_1 \stackrel{N \gg 1}{\approx} \begin{cases} N; & \beta > 1 \\ N^{3-2\beta}; & \beta < 1 \end{cases}. \tag{56}$$

Finally we turn to $\Upsilon_0$, which corresponds to terms where none of the four sites are the same. To compute this, we let both the pair of indices run freely and subtract off the contributions coming from the terms where the pairs coincide and that where there is only one common site, which are nothing but $\Upsilon_2$ and $\Upsilon_1$ respectively. Hence

$$\Upsilon_0 = \sum_{j>i} \frac{1}{|i-j|^\beta} \sum_{l>k} \frac{1}{|k-l|^\beta} - \Upsilon_2 - \Upsilon_1 \tag{57}$$

which using the same arguments as for $\Upsilon_2$ can be re-expressed as

$$\Upsilon_0 = \left( \sum_{r=1}^{N-1} \frac{N-r}{r^\beta} \right)^2 - \Upsilon_2 - \Upsilon_1 \overset{N\gg1}{\approx} \begin{cases} N; & \beta > 1 \\ N^{4-2\beta}; & \beta < 1 \end{cases}. \tag{58}$$

Analysing the asymptotic scaling of $\Upsilon_0$, $\Upsilon_1$, and $\Upsilon_2$ with $N$ shows that $\overline{\mathcal{E}^2}$ (Eq. (23)) is dominated by $\Upsilon_2$, Eq. (51), which in turn leads to Eq. (25) for $\overline{\mathcal{E}^2}$ in the thermodynamic limit.

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
