# Peer review of "Self-consistent theory of many-body localisation in a quantum spin chain with long-range interactions"

_SciPost Physics, doi:SciPost Phys. 7, 042 (2019)_

## Round 1 · Referee Report · Anonymous · 2019-5-24

Strengths

- analytical and numerical results
- clear explanations of the predicted phenomena

Weaknesses

- it is hard to be convinced of the validity of the main results

Report

This paper examine the possibility of MBL phases in systems with long range interactions at finite - infinite - temperature. There have been recent works on MBL phases in these systems but at small temperatures, working on Luttinger liquids. Here the authors find that increasing the range of longitudinal interactions in a spin system favour localisation, while increasing the traverse range (hopping terms) aids delocalisation. This behaviour can be somehow expected by general argument. Moreover they find critical value of the power law exponents for the decay of interactions below (and above) that system is always localised or always delocalised. Their findings are shown by numerical analysis of spectral statistics and mean field analytical results.
Few technical questions
- why the expectation value eq 17 for example can be taken only on zero magnetisation states? This is not the case in a infinite temperature state.
- eq 42 does reproduce the mean field result for W_c directly computed in the short range XXZ chain in the limit alpha and beta ->infinity? This would provide a more justification for validity of the method. Moreover it would be good to benchmark the result of the mean field approach with zero disorder cases, where other methods can be used.
- Is there a free fermion description of the Hamiltonian at some particular value of alpha, beta and J_z/J ?

Finally my main concern is the validity of the fully delocalised or fully localised phase. In the discussions there are numerous "appeal to continuity" namely it is assumed that a free system is continuos to an interacting (long ranged) system. There are no reasons to believe this "continuity". It is true that free systems under small disorder localise by Anderson transition but small long range interactions can break localisation even when these are infinitesimal. Is there a more solid argument to exclude such possibility?

Requested changes

See report. Stronger arguments should be given for the transitions at critical alpha and beta.

---

## Round 2 · Referee Report · Anonymous · 2019-9-15

Report
The authors replied to all my questions and fully addressed the points that were not completely clear. Clearly the validity of mean field approach and finite size numerics is always a problematic issue but this study satisfies all scientific criteria. I recommend publication as it stands.

---

## Round 2 · Author Response

We thank you for your editorial recommendation, and the Referee for his/her comments and questions, on our submission entitled "Self-consistent theory of many-body localisation in a quantum spin chain with long-range interactions". We are pleased to see the strengths of the manuscript appreciated by the Referee, namely concrete analytical results corroborated by numerical results, and clear explanations of the phenomena predicted. We would like to resubmit our modified manuscript, where we believe the points raised by the Referee have been addressed and the text suitably edited in the light of the Referee's helpful report. Before we respond in detail to the Referee's specific technical questions, a few general remarks and responses are in order.
The general concern of the Referee relates to the "the validity of the main results". In order to address this, let us re-state the central result of the paper: in a disordered spin-chain with long-ranged interactions, making the longitudinal interactions longer ranged favours many-body localisation (manifest by a parametric lowering of the critical disorder), whereas making the transverse interactions longer ranged has exactly the opposite effect. We present an analytical theory, of a mean-field nature and on the Fock space. The qualitative predictions of the theory are corroborated by the numerics. The numerical results are wholly independent of, and complementary to, the analytical predictions, and that they qualitatively match is we believe a compelling argument for the validity of the main results. Indeed, the Referee also makes a comment concomitant with our results that ``this behaviour can be somehow expected by general argument''. However, our theory goes well beyond the common lore that long-ranged interactions generally disfavour localisation. As such, we believe that a concrete theory highlighting this qualitative phenomenon is of significance.
Let us now comment briefly on the Referee's points about the \textit{critical} values of $\alpha$ and $\beta$. We would like to clarify that the lines $\alpha=1/2$ and $\beta=1/2$ are not critical lines, but simply represent boundaries of no-go regions for the localised or delocalised phases. The mean-field prediction for the critical surface in the $\alpha$-$\beta$-$W$ parameter space is described by Eq. (42) (and illustrated in Fig. 2b). For any particular choice of disorder and interaction strengths, the critical boundaries in the $\alpha-\beta$ plane lie away from these lines (see e.g. panel c1 of Fig. 2). A more precise statement is that the aforementioned lines are limiting values on the critical $\alpha$ and $\beta$ obtained from the relative scaling with system size of effective disorder and connectivities on the Fock space. Of course, these limiting values are from our mean-field treatment of the problem. However, what stands firm is that there exists a region in $\alpha$-$\beta$-$W$ parameter space where the infinite-temperature states are localised for infinitesimal disorder and, similarly, delocalised even for arbitrarily strong disorder. This is consonant with longer-ranged longitudinal and transverse interactions respectively favouring and disfavouring localisation; which is the main result of the work. We have modified the text in the manuscript as well as the caption to Fig. 2 to highlight this clarification.
The Referee also raises questions regarding our ``appeal to continuity'' in the Discussion section (Sec. 6). We point out respectfully that we don't in fact appeal to continuity from any non-interacting model to actually derive our results. Rather, we use it simply to connect to previous results, and to provide an explanation for
the apparent discrepancy at small values of $\beta$ in Fig. 4 between the numerical results and the mean-field predictions. The Referee asks why a free-fermionic Anderson localised system should remain localised upon adding infinitesimally small but long-ranged interactions. This is indeed a non-trivial question, but comes under the very umbrella of questions that our manuscript seeks to answer, namely the phase diagram of the model; which we obtain from both the mean-field analysis and the numerical results. As we have emphasised in the manuscript, both methods used have their pros and cons; while one is an
analytic but approximate calculation in the thermodynamic limit, the other is exact but affected by finite-size effects. Hence, as is often the case in studies of MBL, while the two sets of results agree qualitatively there can be some tension regarding quantitative specifics. In such a case it is natural to invoke arguments based on physical grounds. Let us briefly clarify the argument in this case.
For simplicity, consider here $\alpha\to\infty$ (extension to finite $\alpha$ is elaborated on in our Response below). It is clear that in the limit of $\beta\to 0$, the system is effectively a disordered free-fermionic system with nearest-neighbour hoppings, and hence is Anderson localised with vanishing critical disorder strength (in 1D). This is confirmation of the fact that the apparent finite critical disorder strength in Fig. 4 at the smallest value of $\beta$ is a numerical finite-size effect. Second, the rather good agreement between the two sets of results at $\beta \gtrsim 1$ (Fig. 4) suggests that there does exist a localised phase in this regime, with the critical disorder growing with $\beta$ but saturating to a finite value as $\beta\to\infty$. So we are left with the contentious regime of small but finite $\beta$. Since we find no evidence of any non-monotonicity as a function of disorder strength and $\beta$, one can reasonably speculate only two scenarios: (i) the critical disorder vanishes at a finite value of $\beta$; (ii) the critical disorder vanishes as $\beta \to 0$. The mean-field treatment predicts the former; although the precise value of $\beta$ at which the critical disorder vanishes is an open question that naturally invites future work. We have now significantly edited the text in the Discussion section to clarify these points.
We reply below to the technical questions of the Referee point-wise.
We thank the Referee for his/her comments, which undoubtedly have improved the manuscript. We hope that we have addressed
the points satisfactorily, and that the modified manuscript is suitable for publication in SciPost Physics.
Yours sincerely,
S. Roy and D.E. Logan
* * *
Response to Referee's specific questions
* * *
-- The model considered in the work, described by the Hamiltonian Eq. (1), conserves total magnetisation, $M_z=\sum_\ell\sigma^z_\ell$. Hence, each of the sectors with fixed $M_z$ can be considered independently. We choose to work in the $M_z=0$ sector as it has the largest Fock-space dimension and dominates the entire Fock space made up of all $M_z$ sectors in the thermodynamic limit. We would like to mention that this is very much the conventional case considered in the literature on many-body localisation in systems with conserved magnetisation (or equivalently particle number). While we have not shown it explicitly, we add that the scaling with system size of effective disorder and connectivities on the Fock space, stay the same for any $M_z$ sector whose Fock-space dimension is exponentially large in the system size. We have now mentioned this explicitly right after introducing the model.
-- In the short-ranged limit of $\alpha\to\infty$ and $\beta\to\infty$, the result in Eq. (42) does indeed reproduce the results of the previous mean-field analysis of a short-ranged model presented in Ref. [29]. The critical disorder for the short-ranged case obtained from this mean-field analysis is somewhat lower than that obtained in large-scale exact diagonalisation studies (c.f. Ref [17]), which is quite expected from a mean-field treatment. The quantitative comparison between the mean-field predictions and exact diagonalisation results presented in the present work is in our view remarkably good for intermediate values of $\alpha$ and $\beta$. The reason for this can be argued to be the much higher connectivity (both in real space and Fock space) of the long-ranged models compared to nearest-neighbour models, which makes the former more suitable for a mean-field analysis.
--The model considered here does indeed have a free-fermionic description for
(i) $\alpha\to\infty$ and $J_z=0$ for any value of $\beta$ and $J$
(ii) $\alpha\to\infty$ and $\beta\to0$ for any value of $J_z$ and $J$
Note that the second of the two cases was used to obtain the free-fermionic representation in Eq. (48).
-- Let us now return to the issue of continuity in the phase diagram briefly mentioned before, and elaborate on it with reference to a figure that can be viewed at https://bit.ly/2LOnDvX
We first consider the $\beta=0$ line. In this limit, the longitudinal interaction drops out of the Hamiltonian as a constant owing to magnetisation conservation. On this line, $\alpha\to\infty$ is trivially localised due to Anderson localisation, but that localisation persists for finite values of $\alpha$. This is due to the subtle fact that the effective hoppings, although long-ranged in real space, have no inherent randomness. Such systems are known (see Ref. [63-66]) to exhibit Anderson localisation except for a measure zero set of single-particle eigenstates at the edges of the spectrum. Anderson localisation at $\beta=0$ is hence established and is denoted by the red line in the figure linked above. Second, the rather good match between the mean-field predictions and the numerical results at higher values of $\beta$ (manuscript Fig. 4), suggest that the critical disorder grows with $\beta$, eventually saturating as $\beta\to\infty$ (note that the presence of a many-body localisation transition in nearest-neighbour models is well established). This yields a partial phase boundary shown by the solid black curve. Thus, for small but finite values of $\beta$, the two likely possibilities are that the critical disorder either goes to zero at a finite $\beta$ (the blue dashed line), or it goes to zero only as $\beta\to 0$ (green dotted line). The precise critical value of $\beta$ may or may not be captured accurately by the mean-field treatment and constitutes a topic of future work. However, what still holds is that increasing the range of longitudinal interactions (decreasing $\beta$) favours localisation, as manifest in the decrease of the critical disorder strength.

---

## Round 2 · List of Changes

-- The text following Eq.(1) has been modified to clarify that total magnetisation is conserved in the model and that we work in the zero-magnetisation sector. Infinite temperature traces are then defined accordingly.
-- The caption to Figure. 2 and the pertinent text towards the end of Section. 4 has been edited to better explain the meaning of the $\alpha=1/2$ and $\beta=1/2$ lines in the phase diagram. Emphasis has been laid on clarifying that the lines do not represent critical lines, and boundaries of no-go regions for localised or delocalised phases is an accurate description of them.
-- Section 6 (Discussion) has been substantially revised in the light of the Referees's concerns about our appeal to continuity, and his/her question about why localisation is expected to persist upon addition of weak long-ranged interactions to an Anderson localised system. In particular, the text has been modified
(i) to clarify that the non-interacting fermion limit of the problem has long-ranged but "non-random" hoppings, and its consequences have been discussed
(ii) to explain better why the $\beta=0$ line is localised for all $\alpha$ and $W$ in 1D.
(iii) to add a new discussion on various quantitatively different scenarios that can arise in the $\beta$-$W$ plane, and how they are the same qualitatively.
-- References have been updated, and a couple of new references have been added.

---

## Editorial Decision

published